# PCF-GAN: generating sequential data via the characteristic function of measures on the path space

**Hang Lou**
Department of Mathematics
University College London
hang.lou.19@ucl.ac.uk

**Siran Li**
Department of Mathematics
Shanghai Jiao Tong University
sl4025@nyu.edu

**Hao Ni**
Department of Mathematics
University College London
h.ni@ucl.ac.uk

## Abstract

Generating high-fidelity time series data using generative adversarial networks (GANs) remains a challenging task, as it is difficult to capture the temporal dependence of joint probability distributions induced by time-series data. Towards this goal, a key step is the development of an effective discriminator to distinguish between time series distributions. We propose the so-called PCF-GAN, a novel GAN that incorporates the path characteristic function (PCF) as the principled representation of time series distribution into the discriminator to enhance its generative performance. On the one hand, we establish theoretical foundations of the PCF distance by proving its characteristicity, boundedness, differentiability with respect to generator parameters, and weak continuity, which ensure the stability and feasibility of training the PCF-GAN. On the other hand, we design efficient initialisation and optimisation schemes for PCFs to strengthen the discriminative power and accelerate training efficiency. To further boost the capabilities of complex time series generation, we integrate the auto-encoder structure via sequential embedding into the PCF-GAN, which provides additional reconstruction functionality. Extensive numerical experiments on various datasets demonstrate the consistently superior performance of PCF-GAN over state-of-the-art baselines, in both generation and reconstruction quality.

## 1 Introduction

Generative Adversarial Networks (GANs) have been a powerful tool for generating complex data distributions, *e.g.*, image data. The original GAN suffers from optimisation instability and mode collapse, partially remedied later by an alternative training scheme using *integral probability metric* (IPM) in lieu of Jensen–Shannon divergence. The IPMs, *e.g.*, metrics based on Wasserstein distances or Maximum Mean Discrepancy (MMD), consistently yield good measures between generated and real data distributions, thus resulting in more powerful GANs on empirical data ([14, 2, 24]).

More recently, [1] proposed an IPM based on the characteristic function (CF) of measures on $\mathbb{R}^d$, which has the characteristic property, boundedness, and differentiability. Such properties enable the GAN constructed using this IPM as discriminator ("CF-GAN") to stabilise training and improve generative performance. However, ineffective in capturing the temporal dependency of sequential data, such CF-metric fails to address high-frequency cases due to the curse of dimensionality. To tackle this issue, we take the continuous time perspective of time series and lift discrete time series to the path space ([28, 29, 23]). This allows us to treat time series of variable length, unequal sampling, and high frequency in a unified approach. We propose a *path characteristic function (PCF)* distance to characterise distributions on the path space, and propose the corresponding PCF distance as a novel IPM to quantify the distance between measures on the path space.

37th Conference on Neural Information Processing Systems (NeurIPS 2023).

Built on top of the unitary feature of paths ([26]), our proposed PCF has theoretical foundations deeply rooted in the rough path theory ([7]), which exploits the non-commutativity and the group structure of the unitary feature to encode information on order of paths. The CF may be regarded as the special case of PCF with linear random path and $1 \times 1$ unitary matrix. We show that the *PCF distance* (PCFD) possesses favourable analytic properties, including boundedness and differentiability in model parameters, and we establish the linkages between PCFD and MMD. These results vastly generalise classical theorems on measures on $\mathbb{R}^d$ ([1]), with much more technically involved proofs due to the infinite-dimensionality of path space.

On the numerical side, we design an efficient algorithm which, by optimising the trainable parameters of PCFD, maximises the discriminative power and improves the stability and efficiency of GAN training. Inspired by [25, 41], we integrate the proposed PCF into the IPM-GAN framework, utilising an auto-encoder architecture specifically tailored to sequential data. This model design enables our algorithm to generate and reconstruct realistic time series simultaneously, which has advantages in diverse applications, including privacy preservation ([35]) and semantic representation extraction for downstream tasks ([10]). To assess the efficacy of our PCF-GAN, we conduct extensive numerical experiments on several standard time series benchmarking datasets for both generation and reconstruction tasks.

We summarize key contributions of this work below:

- proposing a new metric for the distributions on the path space via PCF;
- providing theoretical proofs for analytic properties of the proposed loss metric which benefit GAN training;
- introducing a novel PCF-GAN to generate & reconstruct time series simultaneously; and
- reporting substantial empirical results validating the out-performance of our approach, compared with several state-of-the-art GANs with different loss functions on various time series generation and reconstruction tasks.

**Related work**. Given the wide practical use of, and challenges for, realistic time series synthesis ([3, 4]), various approaches are proposed to improve the quality of GANs for synthetic time series generation. Several works, *e.g.*, [43, 45, 36], are devoted to improving the discriminator of GANs to be better suited to distributions induced by time series. Among them, COT-GAN in [43] shares a similar philosophy with PCF-GAN by introducing a novel discriminator based on causal optimal transport (which can be seen as an improved variant of the Sinkhorn divergence tailored to sequential data), while TimeGAN ([45]) shares a similar auto-encoder structure, which improves the generator's quality and enables time series reconstruction. Unlike PCF-GAN, the reconstruction and generation modules of TimeGAN are separated, whereas it has additional stepwise supervised loss and discriminative loss. In a different vein, CEGEN[36], GT-GAN [17], COSCI-GAN [39], and EWGAN[37] focus primarily on the design of network framework and generator architecture, which achieve state-of-the-art results on several benchmarking datasets.

## 2 Preliminaries

The characteristic function of a measure on $\mathbb{R}^d$, namely that the Fourier transform, plays a central role in probability theory and analysis. The path characteristic function (PCF) is a natural extension of the characteristic function to the path space.

### 2.1 Characteristic function distance (CFD) between random variables in $\mathbb{R}^d$

Let $X$ be an $\mathbb{R}^d$-valued random variable with the law $\mu = \mathbb{P} \circ X^{-1}$. The characteristic function of $X$, denoted as $\Phi_X : \mathbb{R}^d \to \mathbb{C}$, maps each $\lambda \in \mathbb{R}^d$ to the expectation of its complex unitary transform: $\Phi_X : \lambda \longmapsto \mathbb{E}_{X \sim \mu} \left[ e^{i\langle \lambda, X \rangle} \right]$. Here $U_\lambda : \mathbb{R}^d \to \mathbb{C}, x \mapsto e^{i\langle \lambda, x \rangle}$ is the solution to the linear controlled differential equation:

$$\mathrm{d}U_\lambda(x) = iU_\lambda(x)\langle \lambda, \mathrm{d}x \rangle, \qquad U_\lambda(\mathbf{0}) = 1, \tag{1}$$

where $\mathbf{0}$ is the zero vector in $\mathbb{R}^d$ and $\langle \cdot, \cdot \rangle$ is the Euclidean inner product on $\mathbb{R}^d$.

References [11, 16] studied the squared *characteristic function distance* (CFD) between two $\mathbb{R}^d$-valued random variables $X$ and $Y$ with respect to another probability distribution $\Lambda$ on $\mathbb{R}^d$:

$$\text{CFD}^2_\Lambda(X, Y) = \mathbb{E}_{Z \sim \Lambda}\left[\left|\Phi_X(Z) - \Phi_Y(Z)\right|^2\right]. \qquad (2)$$

It is proved in [25, 1] that if the support of $\Lambda$ is $\mathbb{R}^d$, then $\text{CFD}_\Lambda$ is a distance metric, so that $\text{CFD}^2_\Lambda(X, Y) = 0$ if and only if $X$ and $Y$ have the same distribution. This justifies the usage of $\text{CFD}^2_\Lambda$ as a discriminator for GAN training to learn finite-dimensional random variables from data.

## 2.2 Unitary feature of a path

Let $\text{BV}\left([0, T]; \mathbb{R}^d\right)$ be the space of $\mathbb{R}^d$-valued paths of bounded variation over $[0, T]$. Consider

$$\mathcal{X} := \left\{\bar{\mathbf{x}} : [0, T] \to \mathbb{R}^{d+1} : \bar{\mathbf{x}}(t) = (t, \mathbf{x}(t)) \text{ for } t \in [0, T]; \mathbf{x} \in \text{BV}\left([0, T]; \mathbb{R}^d\right); \mathbf{x}(0) = 0\right\}. \quad (3)$$

For a discrete time series $x = (t_i, x_i)_{i=0}^N$, where $0 = t_0 < t_1 < \cdots < t_N = T$ and $x_i \in \mathbb{R}^d$ ($i \in \{0, \cdots, N\}$), we can embed it into some $\mathbf{x} \in \mathcal{X}$ whose evaluation at $(t_i)_{i=1}^N$ coincides with $x$. This is well suited for sequence-valued data in the high-frequency limit with finer time-discretisation and is often robust in practice ([27, 26]). Such embeddings are not unique. In this work, we adopt the linear interpolation for embedding, following [23, 18, 32].

Let $\mathbb{C}^{m \times m} := \{m \times m \text{ complex matrices}\}$, $I_m$ be the identity matrix, and $*$ be conjugate transpose. Write $U(m)$ and $\mathfrak{u}(m)$ for the Lie group of $m \times m$ unitary matrices and its Lie algebra, resp.:

$$U(m) = \{A \in \mathbb{C}^{m \times m} : A^* A = I_m\}, \qquad \mathfrak{u}(m) := \{A \in \mathbb{C}^{m \times m} : A^* + A = 0\}.$$

**Definition 2.1.** Let $\mathbf{x} \in \text{BV}\left([0, T]; \mathbb{R}^d\right)$ be a continuous path and $M : \mathbb{R}^d \to \mathfrak{u}(m)$ be a linear map. The unitary feature of $\mathbf{x}$ under $M$ is the solution $\mathbf{y} : [0, T] \to U(m)$ to the following equation:

$$d\mathbf{y}_t = \mathbf{y}_t \cdot M(d\mathbf{x}_t), \qquad \mathbf{y}_0 = I_m. \qquad (4)$$

We write $\mathcal{U}_M(\mathbf{x}) := \mathbf{y}_T$, *i.e.*, the endpoint of the solution path.

By a slight abuse of notations, $\mathcal{U}_M(\mathbf{x})$ is also called the unitary feature of $\mathbf{x}$ under $M$. Unitary feature is a special case of the *Cartan/path development*, for which one may consider paths taking values in any Lie group $G$. We take only $G = U(m)$ here; $m \neq d$ in general ([6, 30]).

**Example 2.2.** *For $M \in \mathcal{L}\left(\mathbb{R}^d, \mathfrak{u}(m)\right)$ and $\mathbf{x} \in \text{BV}\left([0, T]; \mathbb{R}^d\right)$ linear, $\mathcal{U}_M(X) = e^{M(\mathbf{x}_T - \mathbf{x}_0)}$. In particular, when $m = 1$, $\mathfrak{u}(1)$ is reduced to $i\mathbb{R}$ and $M(y) = i\langle \lambda_M, y \rangle$ for some $\lambda_M \in \mathbb{R}^d$.*

Motivated by the universality and characteristic property of unitary features ([7], see Appendix A.3), we constructed a unitary layer which transforms any $d$-dimensional time series $x = (x_0, \cdots, x_N)$ to the unitary feature of its piecewise linear interpolation $\mathbf{X}$. It is a special case of the path development layer [26], when Lie algebra is chosen as $\mathfrak{u}(m)$. In fact, the explicit formula holds: $\mathcal{U}_M(\mathbf{X}) = \prod_{i=1}^{N+1} \exp\left(M(\Delta x_i)\right)$, where $\Delta x_i := x_i - x_{i-1}$ and $\exp$ is the matrix exponential.

**Convention 2.3.** *The space $\mathcal{L}\left(\mathbb{R}^d, \mathfrak{u}(m)\right)$ in which $M$ of Eq. (4) resides is isomorphic to $\mathfrak{u}(m)^d$, where $\mathfrak{u}(m)$ is Lie algebra isomorphic to $\mathbb{R}^{\frac{m(m-1)}{2}}$. For each $\theta \in \mathfrak{u}(m)^d$ given by anti-Hermitian matrices $\{\theta^{(i)}\}_{i=1}^d$, a linear map $M$ is uniquely induced: $M(x) = \sum_{i=1}^d \theta^{(i)} \langle x, e_i \rangle, \forall x \in \mathbb{R}^d$.*

# 3 Path characteristic function loss

## 3.1 Path characteristic function (PCF)

The unitary feature of a path $\mathbf{x} \in \mathcal{X}$ plays a role similar to that played by $e^{i\langle x, \lambda \rangle}$ to an $\mathbb{R}^d$-valued random variable. Thus, for a *random* path $\mathbf{X}$, the *expected* unitary feature can be viewed as the characteristic function for measures on the path space ([7]).

**Definition 3.1.** Let $\mathbf{X}$ be an $\mathcal{X}$-valued random variable and $\mathbb{P}_\mathbf{X}$ be its measure. The path characteristic function (PCF) of $\mathbf{X}$ of order $m \in \mathbb{N}$ is the map $\boldsymbol{\Phi}_\mathbf{X}^{(m)} : \mathcal{L}\left(\mathbb{R}^d, \mathfrak{u}(m)\right) \to \mathbb{C}^{m \times m}$ given by

$$\boldsymbol{\Phi}_\mathbf{X}(M) := \mathbb{E}[\mathcal{U}_M(\mathbf{X})] = \int_\mathcal{X} \mathcal{U}_M(\mathbf{x}) \, d\mathbb{P}_\mathbf{X}(\mathbf{x}).$$

The path characteristic function (PCF) $\Phi_{\mathbf{X}} : \bigoplus_{m=0}^{\infty} \mathcal{L}\left(\mathbb{R}^d, \mathfrak{u}(m)\right) \to \bigoplus_{m=0}^{\infty} \mathbb{C}^{m \times m}$ is defined by the natural grading: $\Phi_{\mathbf{X}}\big|_{\mathcal{L}(\mathbb{R}^d, \mathfrak{u}(m))} = \Phi_{\mathbf{X}}^{(m)}$ for each $m \in \mathbb{N}$.

In the above, $\mathcal{U}_M(\mathbf{x}) \in U(m)$ is the unitary feature of the path $\mathbf{x}$ under $M$. See Definition 2.1.

Similarly to the characteristic function of $\mathbb{R}^d$-valued random variables, the PCF always exists. Moreover, we have the following important result, whose proof is presented in Appendix A.

**Theorem 3.2** (Characteristicity). *Let $\mathbf{X}$ and $\mathbf{Y}$ be $\mathcal{X}$-valued random variables. They have the same distribution (denoted as $\mathbf{X} \overset{d}{=} \mathbf{Y}$) if and only if $\Phi_{\mathbf{X}} = \Phi_{\mathbf{Y}}$.*

### 3.2 A new distance measure via PCF

We now introduce a novel and natural distance metric, which measures the discrepancy between distributions on the path space via comparing their PCFs. Throughout, $d_{\mathrm{HS}}$ denotes the metric associated with the Hilbert–Schmidt norm $\| \bullet \|_{\mathrm{HS}}$ on $\mathbb{C}^{m \times m}$:

$$d_{\mathrm{HS}}(A, B) := \sqrt{\|A - B\|_{\mathrm{HS}}^2} = \sqrt{\mathrm{tr}\left[(A - B)(A - B)^*\right]}.$$

**Definition 3.3.** Let $\mathbf{X}, \mathbf{Y} : [0, T] \to \mathbb{R}^d$ be stochastic processes and $\mathbb{P}_{\mathcal{M}}$ be a probability distribution on $\mathfrak{u}(m)^d := \mathcal{L}\left(\mathbb{R}^d, \mathfrak{u}(m)\right)$ (recall Convention 2.3). Define the squared PCF-based distance (PCFD) between $\mathbf{X}$ and $\mathbf{Y}$ with respect to $\mathbb{P}_{\mathcal{M}}$ as

$$\mathrm{PCFD}_{\mathcal{M}}^2(\mathbf{X}, \mathbf{Y}) = \mathbb{E}_{M \sim \mathbb{P}_{\mathcal{M}}}\left[d_{\mathrm{HS}}^2\left(\Phi_{\mathbf{X}}(M), \Phi_{\mathbf{Y}}(M)\right)\right]. \tag{5}$$

We shall not distinguish between $\mathcal{M}$ and $\mathbb{P}_{\mathcal{M}}$ for simplicity.

PCFD exhibits several mathematical properties, which provide the theoretical justification for its efficacy as the discriminator on the space of measures on the path space, leading to empirical performance boost. First, PCFD has the characteristic property.

**Lemma 3.4** (Separation of points). *Let $\mathbf{X}, \mathbf{Y} \in \mathcal{P}(\mathcal{X})$ and $\mathbf{X} \neq \mathbf{Y}$. Then there exists $m \in \mathbb{N}$, such that if $\mathcal{M}$ is a $\mathfrak{u}(m)^d$-valued random variable with full support, then $PCFD_{\mathcal{M}}(\mathbf{X}, \mathbf{Y}) \neq 0$.*

Furthermore, $\mathrm{PCFD}_{\mathcal{M}}$ has a simple uniform upper bound for any fixed $m \in \mathbb{N}$:

**Lemma 3.5.** *Let $\mathcal{M}$ be a $\mathfrak{u}(m)^d$-valued random variable. Then, for any $\mathrm{BV}\left([0, T]; \mathbb{R}^d\right)$-valued random variables $\mathbf{X}$ and $\mathbf{Y}$, it holds that $\mathrm{PCFD}_{\mathcal{M}}^2(\mathbf{X}, \mathbf{Y}) \leq 2m^2$.*

Under mild conditions, PCFD is *a.e.* differentiable with respect to a continuous parameter, thus ensuring the feasibility of gradient descent in training.

**Theorem 3.6** (Lipschitz dependence on continuous parameter). *Let $\mathcal{X}$ and $\mathcal{Z}$ be subsets of $\mathrm{BV}\left([0, T]; \mathbb{R}^d\right)$, $(\Theta, \rho)$ be a metric space, $\mathbb{Q}$ be a Borel probability measure on $\mathcal{Z}$, and $\mathcal{M}$ be a Borel probability measure on $\mathfrak{u}(m)^d$. Assume that $g : \Theta \times \mathcal{Z} \to \mathcal{X}$, $(\theta, \mathbf{Z}) \mapsto g_\theta(\mathbf{Z})$ is Lipschitz in $\theta$ such that $\mathrm{Tot.Var.}\left[g_\theta(\mathbf{Z}) - g_{\theta'}(\mathbf{Z})\right] \leq \omega(\mathbf{Z})\rho(\theta, \theta')$. In addition, suppose that $\mathbb{E}_{M \sim \mathbb{P}_{\mathcal{M}}}\left[\||M|\|^2\right] < \infty$ and $\mathbb{E}_{\mathbf{Z} \sim \mathbb{Q}}\left[\omega(\mathbf{Z})\right] < \infty$. Then $\mathrm{PCFD}_{\mathcal{M}}\left(g_\theta(\mathbf{Z}), \mathbf{X}\right)$ is Lipschitz in $\theta$. Moreover, it holds that*

$$\left|\mathrm{PCFD}_{\mathcal{M}}\left(g_\theta(\mathbf{Z}), \mathbf{X}\right) - \mathrm{PCFD}_{\mathcal{M}}\left(g_{\theta'}(\mathbf{Z}), \mathbf{X}\right)\right| \leq \sqrt{\mathbb{E}_{M \sim \mathbb{P}_{\mathcal{M}}}\left[\||M|\|^2\right]} \mathbb{E}_{\mathbf{Z} \sim \mathbb{Q}}\left[\omega(\mathbf{Z})\right] \rho(\theta, \theta')$$

*for any $\theta, \theta' \in \Theta$, $\mathbf{Z} \in \mathcal{Z}$, $\mathbf{X} \in \mathcal{X}$, and $\mathcal{M} \in \mathcal{P}\left(\mathfrak{u}(m)^d\right)$.*

*Remark* 3.7. The parameter space $(\Theta, \rho)$ is usually taken to be $\mathbb{R}^{\bar{d}}$ for some $\bar{d} \in \mathbb{N}$. In this case, by Rademacher's theorem $\mathrm{PCFD}_{\mathcal{M}}\left(g_\theta(\mathbf{Z}), \mathbf{X}\right)$ is *a.e.* differentiable in $\theta$.

Similarly to metrics on measures over $\mathbb{R}^d$ (*cf.* [2, 24]), we construct a metric based on PCFD, denoted as $\widetilde{\mathrm{PCFD}}$, on the space $\mathcal{P}(\mathcal{X})$ of Borel probability measures over the path space, and we prove that it metrises the weak-star topology on $\mathcal{P}(\mathcal{X})$. Throughout, $\overset{d}{\to}$ denotes the convergence in law.

**Theorem 3.8** (Informal, convergence in law). *Let $\{\mathbf{X}_n\}_{n \in \mathbb{N}}$ and $\mathbf{X}$ be $\mathcal{X}$-valued random variables with measures supported in a compact subset of $\mathcal{X}$. Then $\widetilde{\mathrm{PCFD}}(\mathbf{X}_n, \mathbf{X}) \to 0 \iff \mathbf{X}_n \overset{d}{\to} \mathbf{X}$.*

The formal statement and proof can be found in Lemma B.2 and Theorem B.8 in the Appendix.

Similar to [40] for $\mathbb{R}^d$, we prove that PCFD can be interpreted as an MMD with a specific kernel $\kappa$ (see Appendix B.3). Example B.12 illustrates that the PCFD has the superior test power for hypothesis testing on stochastic processes compared with CF distance on the flattened time series.

## 3.3 Computing PCFD under empirical measures

Now, we shall illustrate how to compute the PCFD on the path space.

Let $\bar{\mathbf{X}} := \{\mathbf{x}^i\}_{i=1}^n$ and $\bar{\mathbf{Y}} := \{\mathbf{y}^i\}_{i=1}^{n'}$ be i.i.d. drawn respectively from $\mathcal{X}$-valued random variables $\mathbf{X}$ and $\mathbf{Y}$. First, for any linear map $M \in \mathfrak{u}(m)^d$, the empirical estimator of $\mathbf{\Phi}_{\mathbf{X}}(M)$ is the average of unitary features of all observations $\bar{\mathbf{X}} = \{\mathbf{x}_i\}_{i=1}^n$, i.e., $\mathbf{\Phi}_{\bar{\mathbf{X}}}(M) = \frac{1}{n}\sum_{i=1}^n \mathcal{U}_M(\mathbf{x}_i)$. We then parameterise the $\mathfrak{u}(m)^d$-valued random variable $\mathcal{M}$ via the empirical measure $\mathcal{M}_{\theta_M}$, i.e., $\mathcal{M}_{\theta_M} = \sum_{i=1}^k \delta_{M_i}$, where $\theta_M := \{M_i\}_{i=1}^k \in \mathfrak{u}(m)^{d \times k}$ are the trainable model parameters. Finally, define the corresponding *empirical path characteristic function distance* (EPCFD) as

$$\mathrm{EPCFD}_{\theta_M}\left(\bar{\mathbf{X}}, \bar{\mathbf{Y}}\right) = \sqrt{\frac{1}{k}\sum_{i=1}^k \|\mathbf{\Phi}_{\bar{\mathbf{X}}}(M_i) - \mathbf{\Phi}_{\bar{\mathbf{Y}}}(M_i)\|_{\mathrm{HS}}^2}. \tag{6}$$

Our approach to approximating $\mathcal{M}$ via the empirical distribution differs from that in [25], where $\mathcal{M}$ is parameterised by mixture of Gaussian distributions. In §4.1 and §5, it is shown that, by optimising the empirical distribution, a moderately sized $k$ is sufficient for achieving superior performance, in contrast to a larger sample size required by [25].

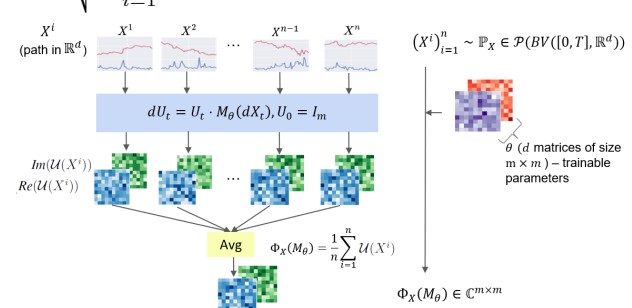

Figure 1: Flowchart of calculating the PCF $\mathbf{\Phi}_{\mathbf{X}}(M_\theta)$.

## 4 PCF-GAN for time series generation

### 4.1 Training of the EPCFD

In this subsection, we apply the EPCFD to GAN training for time series generation as the discriminator. We train the generator to minimise the EPCFD between true and synthetic data distribution, whereas the empirical distribution of $\mathcal{M}$ characterised by $\theta_M \in \mathfrak{u}(m)^{d \times k}$ is optimised by maximising EPCFD.

By an abuse of notation, let $\mathcal{X} := \mathbb{R}^{d \times n_T}$ ($\mathcal{Z} := \mathbb{R}^{e \times n_T}$, resp.) denote the data (noise, resp.) space, composed of $\mathbb{R}^d$ ($\mathbb{R}^e$, resp.) time series of length $n_T$. As discussed in §2.2, $\mathcal{X}$ and $\mathcal{Z}$ can be viewed as path spaces via linear interpolation. Like the standard GANs, our model is comprised of a generator $G_{\theta_g} : \mathcal{Z} \to \mathbb{R}^{d \times n_T}$ and the discriminator $\mathrm{EPCFD}_{\theta_M} : \mathbb{P}(\mathcal{X}) \times \mathbb{P}(\mathcal{X}) \to \mathbb{R}^+$, where $\theta_M \in \mathfrak{u}(m)^{k \times d}$ is the model parameter of the discriminator, which fully characterises the empirical measure of $\mathcal{M}$. The pre-specified noise random variable $\mathbf{Z} = (Z_{t_i})_{i=0}^{n_T - 1}$ is the discretised Brownian motion on $[0, 1]$ with time mesh $\frac{1}{n_T}$. The induced distribution of the fake data is given by $G_{\theta_g}(\mathbf{Z})$. Hence, the min-max objective of our basic version PCF-GAN is

$$\min_{\theta_g} \max_{\theta_M} \mathrm{EPCFD}_{\theta_M}(G_{\theta_g}(\mathbf{Z}), \mathbf{X}).$$

We apply mini-batch gradient descent to optimise the model parameters of the generator and discriminator in an alternative manner. In particular, to compute gradients of the discriminator parameter $\theta_M$, we use the efficient backpropagation algorithm through time introduced in [26], which effectively leverages the Lie group-valued outputs and the recurrence structure of the unitary feature. The initialisation of $\theta_M$ for the optimisation is outlined in the Appendix B.4.1.

**Learning time-dependent Ornstein–Uhlenbeck process**   Following [19], we apply the proposed PCF-GAN to the toy example of learning the distribution of synthetic time series data simulated via the time-dependent Ornstein–Uhlenbeck (OU) process. Let $(\mathbf{X}_t)_{t \in [0,T]}$ be an $\mathbb{R}$-valued stochastic process described by the SDE, i.e., $d\mathbf{X}_t = (\mu t - \theta \mathbf{X}_t)\, dt + \sigma d\mathbf{B_t}$ with $\mathbf{X_0} \sim \mathcal{N}(\mathbf{0}, \mathbf{1})$, where $(\mathbf{B_t})_{\mathbf{t} \in [\mathbf{0}, \mathbf{T}]}$ is 1D Brownian motion and $\mathcal{N}(0, 1)$ is the standard normal distribution. We set $\mu = 0.01$, $\theta = 0.02$, $\sigma = 0.4$ and time discretisation $\delta t = 0.1$. We generate 10000 samples from $t = 0$ to $t = 63$, down-sampled at each integer time point. Figure 2 shows that the synthetic data generated by our GAN model, which uses the EPCFD discriminator, is visually indistinguishable from true data. Also, our model accurately captures the marginal distribution at various time points.

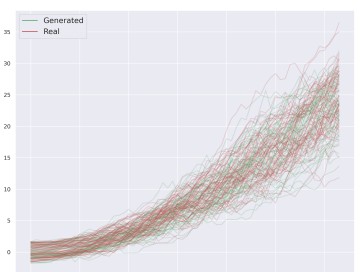 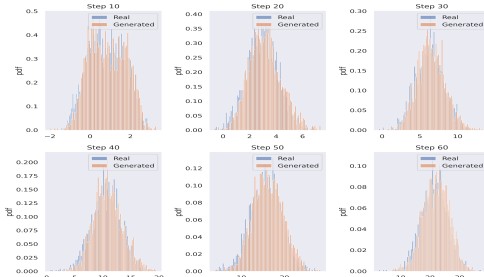

Figure 2: **Left**: Sample paths generated from the time-dependent OU process and synthetic paths from PCF-GAN. **Right**: The marginal distribution comparison at $t \in \{10, 20, 30, 40, 50, 60\}$.

## 4.2 PCF-GAN: learning with PCFD and sequential embedding

In order to effectively learn the distribution of high-dimensional or complex time series, using solely the EPCF loss as the GAN discriminator fails to be the best approach, due to the computational limitations imposed by the sample size $k$ and the order $m$ of EPCFD. To overcome this issue, we adopt the approach [41, 25], and train a generator that matches the distribution of the *embedding* of time series via the auto-encoder structure. Figure 3 illustrates the mechanics of our model.

To proceed, let us first recall the generator $G_{\theta_g} : \mathcal{Z} \to \mathcal{X}$ and introduce the embedding layer $F_{\theta_f}$, which maps $\mathcal{X}$ to $\mathcal{Z}$ (the noise space). Here $\theta_f$ is the model parameters of the embedding layer and will be learned from data. To this end, it is natural to optimize the model parameters $\theta_g$ of the generator by minimising the generative loss $L_{\text{generator}}$, which is the EPCFD distance of the embedding between true distribution $\mathbf{X}$ and synthetic distribution $G_{\theta_g}(\mathbf{Z})$; in formula,

$$L_{\text{generator}}(\theta_g, \theta_M, \theta_f) = \text{EPCFD}_{\theta_M}\left(F_{\theta_f}(G_{\theta_g}(\mathbf{Z})), F_{\theta_f}(\mathbf{X}))\right). \tag{7}$$

**Encoder**$(F_{\theta_f})$-**decoder**$(G_{\theta_g})$ **structure**: The motivation to consider the auto-encoder structure is based on the observation that the embedding might be degenerated when optimizing $L_{\text{generator}}$. For example, no matter whether true and synthetic distribution agrees or not, $F_{\theta_f}$ could be simply a constant function to achieve the perfect generator loss 0. Such a degeneracy can be prohibited if $F_{\theta_f}$ is injective. In heuristic terms, the "good" embedding should capture essential

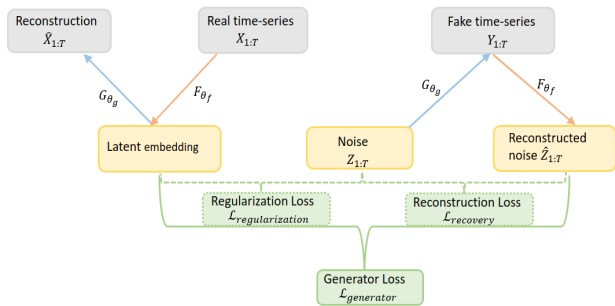

Figure 3: Visualization of the PCF-GAN architecture

information about real time series of $\mathbf{X}$ and allows the reconstruction of time series $\mathbf{X}$ from its embedding $F_{\theta_f}(\mathbf{X})$. This motivates us to train the embedding $F_{\theta_f}$ such that $F_{\theta_f} \circ G_{\theta_g}$ is close to the identity map. If this condition is satisfied, it implies that $F_{\theta_f}$ and $G_{\theta_g}$ are pseudo-inverses of each other, thereby ensuring the desired injectivity. In this way, $F_{\theta_f}$ and $G_{\theta_g}$ serve as the encoder and decoder of raw data, respectively.

To impose the injectivity of $F_{\theta_f}$, we consider two additional loss functions for training $\theta_f$ as follows:

**Reconstruction loss $L_{\text{recovery}}$**: It is defined as the $l^2$ samplewise distance between the original and reconstructed noise by $F_{\theta_f} \circ G_{\theta_g}$, i.e., $L_{\text{recovery}} = \mathbb{E}[|Z - F_{\theta_f}(G_{\theta_g}(\mathbf{Z}))|^2]$. Note that $L_{\text{recovery}} = 0$ implies that $F_{\theta_f}(G_{\theta_g}(\mathbf{z})) = \mathbf{z}$, for any sample $\mathbf{z}$ in the support of $\mathbf{Z}$ almost surely.

**Regularization loss $L_{\text{regularization}}$**: It is proposed to match the distribution of the original noise variable $\mathbf{Z}$ and embedding of true distribution $\mathbf{X}$. It is motivated by the observation that if the perfect generator $G_\theta(\mathbf{Z}) = \mathbf{X}$ and $F_{\theta_f} \circ G_{\theta_g}$ is the identity map, then $\mathbf{Z} = F_{\theta_f}(\mathbf{X})$. Specifically,

$$L_{\text{regularization}} = \text{EPCFD}_{\theta'_M}(\mathbf{Z}, F_{\theta_f}(\mathbf{X})), \tag{8}$$

where we distinguish $\theta'_M$ from $\theta_M$ in $L_{\text{generator}}$. The regularization loss effectively stabilises the training and resolves the mode collapse [41] due to the lack of infectivity of the embedding.

**Training the embedding parameters $\theta_f$**: The embedding layer $F_{\theta_f}$ aims to not only discriminate the real and fake data distributions as a critic, but also preserve injectivity. Hence we optimise the

embedding parameter $\theta_f$ by the following hybrid loss function:

$$\max_{\theta_f} \left( L_{\text{generator}} - \lambda_1 L_{\text{recovery}} - \lambda_2 L_{\text{regularization}} \right), \tag{9}$$

where $\lambda_1$ and $\lambda_2$ are hyper-parameters that balance the three losses.

**Training the EPCFD parameters** $(\theta_M, \theta'_M)$: Note that $L_{\text{generator}}$ and $L_{\text{regularization}}$ have trainable parameters of EPCFD, i.e., $\theta_M$ and $\theta'_M$. Similar to the basic PCF-GAN, we optimize $\theta_M$ and $\theta'_M$ by maximising the EPCFD to improve the discriminative power.

$$\max_{\theta_M} L_{\text{generator}}, \quad \max_{\theta'_M} L_{\text{regularization}} \tag{10}$$

By doing so, we enhance the discriminative power of $\text{EPCFD}_{\theta_M}$ and $\text{EPCFD}_{\theta'_M}$. Consequently, this facilitates the training of the generator such that the embedding of the true data aligns with both the noise distribution and the reconstructed noise distribution.

Differentiability of EPCFD with respect to parameters of the embedding layer and generators are guaranteed by Theorem 3.6, as long as $F_{\theta_f} \circ G_{\theta_g}$ satisfies the Lipschitz condition thereof. Let us also stress on two key advantages of our proposed PCF-GAN. First, it possesses the ability to generate synthetic time series with reconstruction functionality, thanks to the auto-encoder structure in PCF-GAN. Second, by virtue of the uniform boundedness of PCFD shown in Lemma 3.5, our PCF-GAN does not require any additional gradient constraints of the embedding layer and EPCFD parameters, in contrast to other MMD-based GANs and Wasserstein-GAN. It helps with the training efficiency and alleviates the vanishing gradient problem in training sequential networks like RNNs.

We provide the pseudo-code for the proposed PCF-GAN in Algorithm 1.

---

**Algorithm 1** PCF-GAN.

---

1: **Input:** $\mathbb{P}_d$ (real time series distribution), $\mathbb{P}_z$ (noise distribution), $\theta_M, \theta'_M, \theta_f, \theta_g$ (model parameters for EPCFD, critic $F$ and generator $G$), $\lambda_1, \lambda_2 \in \mathbb{R}^+$ (penalty weights), $b$ (batch size), $\eta \in \mathbb{R}$ (learning rate), $n_c$ the iteration number of discriminator per generator update, .
2: **while** $\theta_M, \theta'_M, \theta_M, \theta_c, \theta_g$ not converge **do**
3:      **for** $i \in \{1, \ldots, n_c\}$ **do**
4:          **# train the unitary linear maps in EPCFD**
5:          Sample from distributions: $X \sim \mathbb{P}_d, Z \sim \mathbb{P}_z$.
6:          Generator Loss: $L_{generator} = \text{EPCFD}_{\theta_M}(F_{\theta_f}(X), F_{\theta_f}(G_{\theta_g}(Z)))$
7:          Update: $\theta_M \leftarrow \theta_M + \eta \cdot \nabla_{\theta_M} L_{\text{generator}}$
8:          Regularization Loss: $L_{regularization} = \text{EPCFD}_{\theta'_M}(Z, F_{\theta_f}(X))$
9:          Update: $\theta'_M \leftarrow \theta'_M + \eta \cdot \nabla_{\theta'_M}(L_{\text{regularization}})$
10:          **# train the embedding**
11:          Reconstruction Loss: $L_{\text{recovery}} = \mathbb{E}[|Z - F_{\theta_f}(G_{\theta_g}(Z))|^2]$
12:          Loss on critic: $L_c = L_{\text{generator}} - \lambda_1 \cdot L_{\text{recovery}} - \lambda_2 \cdot L_{\text{regularization}}$
13:          Update: $\theta_f \leftarrow \theta_f + \eta \cdot \nabla_{\theta_c} L_c$
14:      **end for**
15:      **# train the generator**
16:      Sample from distributions: $X \sim \mathbb{P}_d, Z \sim \mathbb{P}_z$.
17:      Generator Loss: $L_{\text{generator}} = \text{EPCFD}_{\mathcal{M}}(F_{\theta_f}(X), F_{\theta_f}(G_{\theta_g}(Z)))$
18:      Update: $\theta_g \leftarrow \theta_g - \eta \cdot \nabla_{\theta_g} L_g$
19: **end while**

---

## 5 Numerical Experiments

To validate its efficacy, we apply our proposed PCF-GAN to a broad range of time series data and benchmark with state-of-the-art GANs for time series generation using various test metrics. Full details on numerics (dataset, evaluation metrics, and hyperparameter choices) are in Appendix C. Additional ablation studies and visualisations of generated samples are reported in Appendix D.

**Baselines**: We take Recurrent GAN (RGAN)[12], TimeGAN [45], and COT-GAN [43] as benchmarking models. These are representatives of GANs exhibiting strong empirical performance for time series generation. For fairness, we compare our model to the baselines while fixing the generators and embedding/discriminator to be the common sequential neural network (2 layers of LSTMs).

**Dataset**: We benchmark our model on four different time series datasets with various characteristics: dimensions, sample frequency, periodicity, noise level, and correlation. **(1) Rough Volatility**: High-frequency synthetic time series data with low noise-to-signal. **(2) Stock**: The daily historical data on ten publicly traded stocks from 2013 to 2021, including as features the volume and high, low, opening, closing, and adjusted closing prices. **(3) Beijing Air Quality** [47]: An UCI multivariate time series on hourly air pollutants data from different monitoring sites. **(4) EEG Eye State** [38]: An UCI dataset of a high frequency and continuous EEG eye measurement. We summarise the key statistics of the datasets in Table 1.

Table 1: Summuary statistics for four datasets

| Dataset | Dimension | Length | Sample rate | Auto-cor (lag 1) | Auto-cor (lag 5) | Cross-cor |
|---------|-----------|--------|-------------|------------------|------------------|-----------|
| RV | 2 | 200 | - | 0.967 | 0.916 | -0.014 |
| Stock | 5 | 20 | 1day | 0.958 | 0.922 | 0.604 |
| Air | 10 | 24 | 1hour | 0.947 | 0.752 | 0.0487 |
| EEG | 14 | 20 | 8ms | 0.517 | 0.457 | 0.418 |

**Evaluation metrics**: The following three metrics are used to assess the quality of generative models. For time series generation/reconstruction, we compare the true and fake/reconstructed distribution by $G_{\theta_g} \circ F_{\theta_f}$ via the below test metrics. **(1) Discriminative score** [45]: We train a post-hoc classifier to distinguish true and fake data. We report the classification error on the test data. The better generative model yields a lower classification error, as it means that the classifier struggles to differentiate between true and fake data. **(2) Predictive score** [45, 12]: We train a post-hoc sequence-to-sequence regression model to predict the latter part of a time series given the first part from the generated data. We then evaluate and report the mean square error (MSE) on the true time series data. The lower MSE indicates better the generated data can be used to train a predictive model. **(3) Sig-MMD** [9, 42]: We use MMD with the signature feature as a generic metric on time series distribution. Smaller the values, indicating closer the distributions, are better. To compute three evaluation metrics, we randomly generated 10,000 samples of true and synthetic (reconstructed) distribution resp. The mean and standard deviation of each metric based on 10 repeated random sampling are reported.

## 5.1 Time series generation

Table 2 indicates that PCF-GAN consistently outperforms the other baselines across all datasets, as demonstrated by all three test metrics. Specifically, in terms of the discriminative score, PCF-GAN achieves a remarkable performance with values of 0.0108 and 0.0784 on the Rough volatility and Stock datasets, respectively. These values are 61% and 39% lower than those achieved by the second-best model. Regarding the predictive score, PCF-GAN achieves the best result across all four datasets. While COT-GAN surpasses PCF-GAN in terms of the Sig-MMD metric on the EEG dataset, PCF-GAN consistently outperforms the other models in the remaining three datasets. Additionally, to assess the fitting on auto-correlation, cross-correlation and marginal distribution, we include the corresponding numerical results in Table 4 in Appendix D.4. For a qualitative analysis of generative quality, we provide the visualizations of generated samples for all models and datasets in Appendix D without selective bias. Furthermore, to showcase the effectiveness of our auto-encoder architecture for the generation task, we present an ablation study in Appendix D.

Table 2: Performance comparison of PCF-GAN and baselines. Best for each task shown in bold.

| | Task | Generation | | | | Reconstruction | |
|---|---|---|---|---|---|---|---|
| Dataset | Test Metrics | RGAN | COT-GAN | TimeGAN | PCF-GAN | TimeGAN (R) | PCF-GAN(R) |
| RV | Discriminative | .0271±.048 | .0499±.068 | .0327±.019 | **.0108±.006** | .5000±.000 | **.2820±.082** |
| | Predictive | .0393±.000 | .0395±.000 | .0395±.001 | **.0390±.000** | .0590±.003 | **.0398±.001** |
| | Sig-MMD | .0163±.004 | .0116±.003 | .0027±.004 | **.0024±.001** | 3.308±1.34 | **.0960±.050** |
| Stock | Discriminative | .1283±.015 | .4966±.002 | .3286±.063 | **.0784±.028** | .4943±.002 | **.3181±.038** |
| | Predictive | .0132±.000 | .0144±.000 | .0139±.000 | **.0125±.000** | .1180±.012 | **.0127±.000** |
| | Sig-MMD | .0248±.008 | .0029±.000 | .0272±.006 | **.0017±.000** | .7587±.186 | **.0078±.004** |
| Air | Discriminative | .4549±.012 | .4992±.002 | .3460±.025 | **.2326±.058** | .4999±.000 | **.4140±.013** |
| | Predictive | .0261±.001 | .0260±.001 | .0256±.000 | **.0237±.000** | .0619±.004 | **.0289±.000** |
| | Sig-MMD | .0456±.015 | .0128±.002 | .0146±.026 | **.0126±.005** | .4141±.078 | **.0359±.012** |
| EEG | Discriminative | .4908±.003 | .4931±.007 | .4771±.008 | **.3660±.025** | .5000±.000 | **.4959±.003** |
| | Predictive | .0315±.000 | .0304±.000 | .0342±.001 | **.0246±.000** | .0499±.001 | **.0328±.001** |
| | Sig-MMD | .0602±.010 | **.0102±.002** | .0640±.025 | .0180±.004 | .0700±.021 | **.0641±.019** |

## 5.2 Time series reconstruction

As TimeGAN is the only baseline model incorporating reconstruction capability, for reconstruction tasks we only compare with TimeGAN. The reconstructed examples of time series using both PCF-GAN and TimeGAN are shown in Figure 4; see Appendix D for more samples.

Visually, the PCF-GAN achieves better reconstruction results than TimeGAN by producing more accurate reconstructed time series samples. Notably, the reconstructed samples from PCF-GAN preserve the temporal dependency of original time series for all four datasets, while some reconstructed samples from TimeGAN in EEG and Stock datasets are completely mismatched. This is further quantified in Table 2 on the reconstruction task, where the reconstructed samples from PCF-GAN consistently outperform those from TimeGAN in terms of all test metrics.

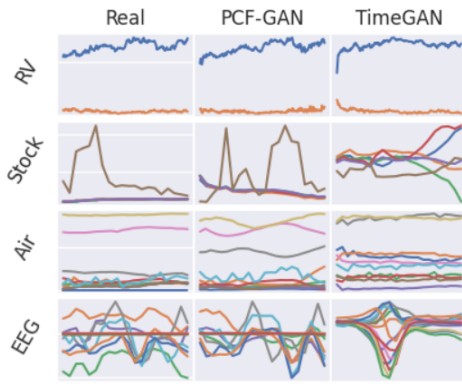

Figure 4: Examples of time series reconstruction using PCF-GAN and TimeGAN.

## 5.3 Training stability and efficiency

Figure 5 demonstrates the training progress of the PCF-GAN on RV dataset. Compared to the fluctuating generator loss typically observed in traditional GANs, the PCF-GAN yields better convergence by leveraging the autoencoder structure. This is achieved by minimising reconstruction and regularisation losses, which ensures the injectivity of $F_{\theta_f}$ and enables production of a semantic embedding throughout the training process. The decay of generator loss in the embedding space directly reflects the improvement in the quality of the generated time series. This is particularly useful for debugging and conducting hyperparameter searches. Furthermore, decay in both recovery and regularisation loss signifies the enhanced performance of the autoencoder.

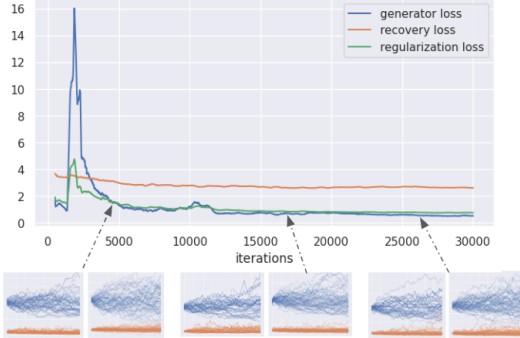

Figure 5: Training curves for PCF-GAN of real (left) and generated (right) time series distributions on the Rough Volatility dataset at different training iterations. Plotted by a moving average over a window with 500 iterations.

By leveraging the effective critic $F_{\theta_f}$, we achieve enhanced performance with a moderate increase in parameters (ranging from 1200 to 6400) within $\theta_M$ of EPCFD. The training of these additional parameters is highly efficient in PCF-GAN, while still outperforming all baseline models. Specifically, our algorithm is approximately twice as fast as TimeGAN (using three extra critic modules) and three times as fast as COT-GAN (with one additional critic module and the Sinkhorn algorithm). However, it takes 1.5 times as long as RGAN due to the extra training required on $\theta_M$.

## 6 Conclusion & Broader impact

**Conclusion** We introduce a novel, principled and efficient PCF-GAN model based on PCF for generating high-fidelity sequential data. With theoretical support, it achieves state-of-the-art generative performance with additional reconstruction functionality in various tasks of time series generation.

**Limitation and future work** In this work, we use LSTM-based networks for the autoencoder and do not explore other sequential models (e.g., transformers). The suitable choice of network architecture for the autoencoder may further improve the efficacy of the proposed PCF-GAN on more complicated data, *e.g.*, video and skeletal human action sequence, which merits further investigation. As a distance metric on time series, PCFD can be flexibly incorporated with other advanced generators of time

series GAN models, hence may further improve the performance. For example, one can replace the average cross-entropy loss used in [17, 39] and the Wasserstein distance in [36] by PCFD, with some simple modifications on the discriminators. Furthermore, although we establish the link between PCFD and MMD, it is interesting to design efficient algorithms to compute the kernel specified in Appendix B.3.

**Broader impact** Like other GAN models, this model has the potential to aid data-hungry algorithms by augmenting small datasets. Additionally, it can enable data sharing in domains such as finance and healthcare, where sensitive time series data is plentiful. However, it is important to acknowledge that the generation of synthetic data also carries the risk of potential misuse (*e.g.* generating fake news).

## Acknowledgments and Disclosure of Funding

The research of SL is supported by NSFC (National Natural Science Foundation of China) Grant No. 12201399, and the Shanghai Frontiers Science Center of Modern Analysis. This research project is also supported by SL's visiting scholarship at New York University-Shanghai. HN is supported by the EPSRC under the program grant EP/S026347/1 and The Alan Turing Institute under the EPSRC grant EP/N510129/1. LH is supported by University College London and the China Scholarship Council under the UCL-CSC scholarship (No. 201908060002). SL and HN are supported by the SJTU-UCL joint seed fund WH610160507/067. HN and HL are supported by the Ecosystem Leadership Award under the EPSRC Grant OobfJ22/100020 and The Alan Turing Institute in part. HN is grateful to Jiajie Tao and Zijiu Lyu for proofreading the paper.

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

## A    Preliminaries

### A.1    Paths with bounded variation

**Definition A.1.** Let $X : [0, T] \to \mathbb{R}^d$ be a continuous path. The total variation of $X$ on the interval $[0, T]$ is defined by

$$\text{Tot.Var.}(X) := \sup_{\mathcal{D} \subset [0,T]} \left\{ \sum_\ell \left| X_{t_\ell} - X_{t_{\ell-1}} \right| \right\} \tag{11}$$

where the supremum is taken over all finite partitions $\mathcal{D} = \{t_\ell\}_{\ell=0}^N$ of $[0, T]$. When $\text{Tot.Var.}(X)$ is finite, say that $X$ is a path of bounded variation (BV-path) on $[0, T]$ and denote $X \in \text{BV}\left([0, T]; \mathbb{R}^d\right)$.

BV-paths can be defined without the continuity assumption, but we shall not seek for greater generality in this work. It is well-known that

$$\|X\|_{\text{BV}} := \|X\|_{C^0([0,T])} + \text{Tot.Var.}(X)$$

defines a norm (the BV-norm). There is a more general notion of paths of finite $p$-variation for $p \geq 1$ (see [29]), where the case $p = 1$ corresponds to BV-paths discussed above. We restrict ourselves to $p = 1$, as this is sufficient for the study of sequential data in practice as piecewise linear approximations of continuous paths.

**Definition A.2.** (Concatenation of paths) Let $X : [0, s] \to \mathbb{R}^d$ and $Y : [s, t] \to \mathbb{R}^d$ be two continuous paths. Their concatenation denoted as the path $X \star Y : [0, t] \to \mathbb{R}^d$ is defined by

$$(X \star Y)_u = \begin{cases} X_u, & u \in [0, s], \\ Y_u - Y_s + X_s, & u \in [s, t]. \end{cases}$$

**Definition A.3** (Tree-like equivalence)**.** A continuous path $X : [0, T] \to \mathbb{R}^d$ is called tree-like if there is an $\mathbb{R}$-tree $\mathcal{T}$, a continuous function $\phi : [0, T] \to \mathcal{T}$, and a function $\psi : \mathcal{T} \to \mathbb{R}^d$ such that $\phi(0) = \phi(T)$ and $X = \psi \circ \phi$.

Let $\overleftarrow{X} : [0, T] \to \mathbb{R}^d$ denote the time-reversal of continuous path $X$, namely that $\overleftarrow{X}(t) = X(T - t)$. We say that $X$ and $Y$ are in tree-like equivalence (denoted as $X \sim_\tau Y$) if $X \star \overleftarrow{Y}$ is tree-like.

An important example is when path $X$ is a time re-parameterisation of $Y$. That is, for $X \in \text{BV}\left([0, T]; \mathbb{R}^d\right)$, take a nondecreasing surjection $\lambda : [0, T] \to [T_1, T_2]$, and take $X(t) = Y(\lambda(t))$.

### A.2    Matrix groups and algebras

The unitary group and symplectic group are subsets of the space of $m \times m$ matrices:

$$U(m) := \left\{ A \in \mathbb{C}^{m \times m} : A^* A = A A^* = I_m \right\},$$

$$Sp(2m, \mathbb{C}) := \left\{ A \in \mathbb{C}^{2m \times 2m} : A^* J_m A = J_m \right\}.$$

where $J_m := \begin{pmatrix} 0 & I_m \\ -I_m & 0 \end{pmatrix}$ and $I_m \in \mathbb{C}^{m \times m}$ is the identity. Their corresponding Lie algebras are

$$\mathfrak{u}(m) := \left\{ A \in \mathbb{C}^{m \times m} : A^* + A = 0 \right\},$$

$$\mathfrak{sp}(2m, \mathbb{C}) := \left\{ A \in \mathbb{C}^{2m \times 2m} : A^* J_m + J_m A = 0 \right\}.$$

The unitary group is compact and is a group of isometries of matrix multiplication with respect to the Hilbert–Schmidt norm. Such properties are crucial for establishing theorems and properties related to the path characteristic function (PCF), as discussed in subsequent sections.

The compact symplectic group $\text{Sp}(m)$ is the simply-connected maximal compact real Lie subgroup of $\text{Sp}(2m, \mathbb{C})$. It is the real form of $\text{Sp}(2n, \mathbb{C})$, and satisfies

$$\text{Sp}(m) = \text{Sp}(2m, \mathbb{C}) \cap U(2m).$$

Note that $U(m)$ and $\text{Sp}(m)$ are both *real* Lie groups, albeit they have complex entries in general.

## A.3 Unitary feature of a path

Recall Definition 2.1 for the unitary feature, reproduced below:

**Definition A.4.** Let $M : \mathbb{R}^d \to \mathfrak{u}(m)$ be a linear map and let $X \in \mathrm{BV}\left([0,T];\mathbb{R}^d\right)$ be a BV-path. The unitary feature [*a.k.a.* the path development on the unitary group $U(m)$] of **x** under $M$ is the solution to the equation

$$\mathrm{d}\mathbf{y}_t = \mathbf{y}_t \cdot M(\mathrm{d}\mathbf{x}_t) \qquad \text{for all } t \in [0,T] \text{ with } \mathbf{Y}_0 = I_m.$$

We write $\mathcal{U}_M(\mathbf{x}) := \mathbf{y}_T$.

Definition 2.1 is motivated by [7, §4]. Consider $M \in \mathcal{L}\left(\mathbb{R}^d, \mathfrak{u}(\mathcal{H}_{\mathrm{fd}})\right)$ with $\mathcal{H}_{\mathrm{fd}}$ ranging over all finite-dimensional complex Hilbert spaces. Extend $M$ by naturality to the tensor algebra over $\mathbb{R}^d$; that is, define $\widetilde{M} : T\left((\mathbb{R}^d)\right) \equiv \bigoplus_{k=0}^{\infty} (\mathbb{R}^d)^{\otimes k} \to \mathfrak{u}(\mathcal{H}_{\mathrm{fd}})$ by linearity and the following rule:

$$\widetilde{M}(v_1 \otimes \ldots \otimes v_k) := M(v_1) \ldots M(v_k) \quad \text{for any } k \in \mathbb{N} \text{ and } v_1, \ldots, v_k \in \mathbb{R}^d.$$

Then denote by $\mathcal{A}\left(\mathbb{R}^d\right)$ the totality of such $\widetilde{M}$. Any element in $\mathcal{A}\left(\mathbb{R}^d\right)$ is a unitary representation of the Lie group $\mathcal{G}\left((\mathbb{R}^d)\right) := \left\{\text{group-like elements in } T\left((\mathbb{R}^d)\right)\right\}$. See [7, p.4059].

The following two lemmas are contained in [26].

**Lemma A.5.** *[Multiplicativity] Let $X \in \mathrm{BV}\left([0,s],\mathbb{R}^d\right)$ and $Y \in \mathrm{BV}\left([s,t],\mathbb{R}^d\right)$. Denote by $X * Y$ their concatenation: $(X * Y)(v) = X(v)$ for $v \in [0,s]$ and $Y(v) - Y(s) + X(s)$ for $v \in [s,t]$. Then $\mathcal{U}_M(X * Y) = \mathcal{U}_M(X) \cdot \mathcal{U}_M(Y)$ for all $M \in \mathcal{L}\left(\mathbb{R}^d, \mathfrak{u}(m)\right)$.*

We shall compute by Lemma A.5 and Example 2.2 the unitary feature of piecewise linear paths.

**Lemma A.6** (Invariance under time-reparametrisation). *Let $X \in \mathrm{BV}([0,T],\mathbb{R}^d)$ and let $\lambda : t \mapsto \lambda_t$ be a non-decreasing $\mathcal{C}^1$-diffeomorphism from $[0,T]$ onto $[0,S]$. Define $X_t^\lambda := X_{\lambda_t}$ for $t \in [0,T]$. Then, for all $M \in \mathcal{L}\left(\mathbb{R}^d, \mathfrak{u}(m)\right)$ and for every $s,t \in [0,T]$, it holds that $\mathcal{U}_M\left(X_{\lambda_s,\lambda_t}\right) = \mathcal{U}_M\left(X_{s,t}^\lambda\right)$.*

A key property of the unitary feature is that it completely determines the law of random paths:

**Theorem A.7** (Uniqueness of unitary feature). *For any two paths $\mathbf{X}_1 \neq \mathbf{X}_2$ in $\mathcal{X}$, there exists an $M \in \mathcal{L}\left(\mathbb{R}^d, \mathfrak{u}(m)\right)$ with some $m \in \mathbb{N}$ such that $\mathcal{U}_M(\mathbf{X}_1) \neq \mathcal{U}_M(\mathbf{X}_2)$.*

*Proof.* For $\mathbf{X}_1 \neq \mathbf{X}_2$ in $\mathcal{X}$, by uniqueness of signature over BV-paths (*cf.* [15]) one has $\mathrm{Sig}(\mathbf{X}_1) \neq \mathrm{Sig}(\mathbf{X}_2)$ in $\mathcal{G}\left((\mathbb{R}^d)\right)$. Here we use the fact that the signatures of BV-paths are group-like elements in the tensor algebra. Then, as $\mathcal{A}\left(\mathbb{R}^d\right)$ separates points over $\mathcal{G}\left((\mathbb{R}^d)\right)$ (*cf.* [7, Theorem 4.8]), there is $M \in \mathcal{L}\left(\mathbb{R}^d, \mathfrak{u}(m)\right)$ such that $\widetilde{M}\left[\mathrm{Sig}(\mathbf{X}_1)\right] \neq \widetilde{M}\left[\mathrm{Sig}(\mathbf{X}_2)\right]$; hence $M(\mathbf{X}_1) \neq M(\mathbf{X}_2)$. Therefore, by considering the $U(m)$-valued equation $\mathrm{d}\mathbf{Y}_t = \mathbf{Y}_t \cdot M(\mathrm{d}\mathbf{X}_t)$ with $\mathbf{Y}_0 = I_m$, we conclude that $\mathcal{U}_M(\mathbf{X}_1) \neq \mathcal{U}_M(\mathbf{X}_2)$. $\square$

**Theorem A.8** (Universality of unitary feature). *Let $\mathcal{K} \subset \mathrm{BV}\left([0,T];\mathbb{R}^d\right)$ be a compact subset. For any continuous function $f : \mathcal{K} \to \mathbb{C}$ and any $\epsilon > 0$, there exists an $m_\star \in \mathbb{N}$ and finitely many $M_1, \cdots, M_N \in \mathcal{L}\left(\mathbb{R}^d, \mathfrak{u}(m_\star)\right)$ as well as $L_1, \ldots, L_N \in \mathcal{L}\left(U(m_\star);\mathbb{C}\right)$, such that*

$$\sup_{\mathbf{X} \in \mathcal{K}} \left| f(\mathbf{X}) - \sum_{i=1}^{N} L_i \circ \mathcal{U}_{M_i}(\mathbf{X}) \right| < \epsilon. \tag{12}$$

*Proof.* It follows from [26, Theorem A.4] and the universality of signature in [8] that Eq. (12) holds with $M_j \in \mathcal{L}\left(\mathbb{R}^d, \bigoplus_{m \in \mathbb{N}} \mathfrak{u}(m)\right)$ and $L_j \in \mathcal{L}\left(\bigoplus_{m \in \mathbb{N}} U(m);\mathbb{C}\right)$ and $\epsilon/2$ in place of $\epsilon$. By a simple approximation via restricting the ranges of $M_j$ and domains of $L_j$, we may obtain (without relabelling) $M_j \in \mathcal{L}\left(\mathbb{R}^d, \bigoplus_{m=0}^{m_\star} \mathfrak{u}(m)\right)$ and $L_j \in \mathcal{L}\left(\bigoplus_{m=0}^{m_\star} U(m);\mathbb{C}\right)$ that verify Eq. (12). We conclude by the flag structure of $U(1) \subset U(2) \subset U(3) \subset \ldots$ and $\mathfrak{u}(1) \subset \mathfrak{u}(2) \subset \mathfrak{u}(3) \subset \ldots$. $\square$

# B Path Characteristic loss

## B.1 Path Characteristic function

**Theorem B.1.** *Let* $\mathbf{X}$ *be a* $\mathcal{X}$*-valued random variable with associated probability measure* $\mathbb{P}_{\mathbf{X}}$. *The path characteristic function* $\boldsymbol{\Phi}_{\mathbf{X}}$ *uniquely characterises* $\mathbb{P}_{\mathbf{X}}$.

*Proof.* Assume that $\mathbb{P}_{\mathbf{X}_1} \neq \mathbb{P}_{\mathbf{X}_2}$. Then $\mathrm{Sig}(\mathbf{X}_1) \neq \mathrm{Sig}(\mathbf{X}_2)$ by the uniqueness of signature over BV-paths (*cf.* [15]). It is proved in [26, Lemma 2.6] that $\mathcal{U}_M(\mathbf{X}_i) = \widetilde{M}(\mathrm{Sig}(\mathbf{X}_i))$ for any $M \in \mathcal{L}(\mathbb{R}^d, \mathfrak{u}(m)); i \in \{1,2\}$. Hence $\boldsymbol{\Phi}_{\mathbf{X}_i} = \int_{\mathcal{X}} \widetilde{M}(\mathrm{Sig}(\mathbf{x})) \, \mathrm{d}\mathbb{P}_{\mathbf{X}_i}(\mathbf{x})$. But as in the proof of Theorem A.7, $\mathcal{A}(\mathbb{R}^d)$ separates points over $\mathcal{G}((\mathbb{R}^d))$ (*cf.* [7, Theorem 4.8]) and the signature of BV-paths lies in $\mathcal{G}((\mathbb{R}^d))$. Therefore, there is an $M \in \mathcal{L}(\mathbb{R}^d, \mathfrak{u}(m))$ such that $\boldsymbol{\Phi}_{\mathbf{X}} \neq \boldsymbol{\Phi}_{\mathbf{Y}}$. $\qquad\square$

## B.2 Distance metric via path characteristic function

**Lemma B.2.** $\mathrm{PCFD}_{\mathcal{M}}$ *in Eq. (5) defines a pseudometric on the path space* $\mathcal{X}$ *for any* $m \in \mathbb{N}$ *and* $\mathcal{M} \in \mathcal{P}(\mathcal{L}(\mathbb{R}^d, \mathfrak{u}(m)))$. *In addition, suppose that* $\{\mathcal{M}_j\}_{j\in\mathbb{N}}$ *is a countable dense subset in* $\mathcal{P}(\mathcal{L}(\mathbb{R}^d, \bigoplus_{m\in\mathbb{N}} \mathfrak{u}(m)))$. *Then the following defines a metric on* $\mathcal{X}$:

$$\widetilde{\mathrm{PCFD}}(\mathbf{X}, \mathbf{Y}) := \sum_{j=1}^{\infty} \frac{\min\{1, \mathrm{PCFD}_{\mathcal{M}_j}(\mathbf{X}, \mathbf{Y})\}}{2^j}. \tag{13}$$

In Lemma B.2 above, $\mathcal{L}(\mathbb{R}^d, \bigoplus_{m\in\mathbb{N}} \mathfrak{u}(m)) \cong (\mathbb{R}^d)^* \widehat{\otimes}_\pi (\bigoplus_{m\in\mathbb{N}} \mathfrak{u}(m))$ where $\widehat{\otimes}_\pi$ is the completion of the projective tensor product and $\bigoplus_{m\in\mathbb{N}} \mathfrak{u}(m)$ is a Banach space under the norm $\|T\| := \sum_{m\in\mathbb{N}} \|T^{(m)}\|_{\mathrm{HS}} < \infty$. Here $T^{(m)}$ denotes the $m^{\mathrm{th}}$-projection of $T$ on $\mathfrak{u}(m)$. Therefore, such a sequence $\{\mathcal{M}_j\}_{j\in\mathbb{N}}$ always exists since $\mathcal{P}(\mathcal{L}(\mathbb{R}^d, \bigoplus_{m\in\mathbb{N}} \mathfrak{u}(m)))$, being the space of Borel probability measures over a Polish space, is itself a Polish space. See [33].

*Proof.* Non-negativity, symmetry, and that $\mathrm{PCFD}_{\mathcal{M}}(\mathbf{X}, \mathbf{X}) = 0$ are clear. That $\mathrm{PCFD}_{\mathcal{M}}(\mathbf{X}, \mathbf{Y}) \leq \mathrm{PCFD}_{\mathcal{M}}(\mathbf{X}, \mathbf{Z}) + \mathrm{PCFD}_{\mathcal{M}}(\mathbf{Z}, \mathbf{Y})$ follows from the triangle inequality of the Hilbert–Schmidt norm and the linearity of expectation. This shows that $\mathrm{PCFD}_{\mathcal{M}}$ is a pseudometric for each $\mathcal{M}$.

In addition, $\mathrm{PCFD}_{\mathcal{M}}(\mathbf{X}, \mathbf{Y}) = 0$ implies that

$$\int_{\mathcal{L}(\mathbb{R}^d, \mathfrak{u}(m))} d_{\mathrm{HS}}^2(\boldsymbol{\Phi}_{\mathbf{X}}(M), \boldsymbol{\Phi}_{\mathbf{Y}}(M)) \, \mathrm{d}\mathbb{P}_{\mathcal{M}}(M)$$

$$= \int_{\mathcal{L}(\mathbb{R}^d, \mathfrak{u}(m))} \|\mathbb{E}[\mathcal{U}_M(\mathbf{X})] - \mathbb{E}[\mathcal{U}_M(\mathbf{Y})]\|_{\mathrm{HS}}^2 \, \mathrm{d}\mathbb{P}_{\mathcal{M}}(M) = 0.$$

So, if $\mathbb{P}_{\mathcal{M}}$ is supported on the whole of $\mathcal{L}(\mathbb{R}^d, \mathfrak{u}(m))$, then $\boldsymbol{\Phi}_{\mathbf{X}}(M) = \boldsymbol{\Phi}_{\mathbf{Y}}(M)$ for any $M \sim \mathbb{P}_{\mathcal{M}}$.

Now, by density of $\{\mathcal{M}_j\}_{j\in\mathbb{N}}$ in $\mathcal{P}(\mathcal{L}(\mathbb{R}^d, \bigoplus_{m\in\mathbb{N}} \mathfrak{u}(m)))$, there exists a subsequence $\{\mathcal{M}_{j(m)}\}$ such that $\mathcal{M}_{j(m)}$ has full support on $\mathcal{L}(\mathbb{R}^d, \mathfrak{u}(m))$ for each $m \in \mathbb{N}$. Thus, $\widetilde{\mathrm{PCFD}}(\mathbf{X}, \mathbf{Y}) = 0$ implies that $\boldsymbol{\Phi}_{\mathbf{X}} = \boldsymbol{\Phi}_{\mathbf{Y}}$ on a dense subset of $\mathcal{L}(\mathbb{R}^d, \mathfrak{u}(m))$ *for every* $m \in \mathbb{N}$. We conclude by the characteristicity Theorem 3.2 and a continuity argument. $\qquad\square$

**Lemma B.3** (Lemma 3.5)**.** *Let* $\mathcal{M}$ *be an* $\mathcal{L}(\mathbb{R}^d, \mathfrak{u}(m))$*-valued random variable. Then for any* $\mathrm{BV}([0,T]; \mathbb{R}^d)$*-valued random variables* $\mathbf{X}$ *and* $\mathbf{Y}$, *it holds that*

$$\mathrm{PCFD}_{\mathcal{M}}(\mathbf{X}, \mathbf{Y}) \leq 2m^2.$$

*Proof.* As $(\mathbb{C}^{m\times m}, \|\bullet\|_{\mathrm{HS}})$ is a Hilbert space, from the Pythagorean theorem one deduces that $d_{\mathrm{HS}}^2(\boldsymbol{\Phi}_{\mathbf{X}}(M), \boldsymbol{\Phi}_{\mathbf{Y}}(M)) \leq \|\boldsymbol{\Phi}_{\mathbf{X}}(M)\|_{\mathrm{HS}}^2 + \|\boldsymbol{\Phi}_{\mathbf{Y}}(M)\|_{\mathrm{HS}}^2$. Both $\boldsymbol{\Phi}_{\mathbf{X}}(M), \boldsymbol{\Phi}_{\mathbf{Y}}(M)$ are expectations of $U(m)$-valued random variables, and $\|U\|_{\mathrm{HS}} := \sqrt{\mathrm{tr}(UU^*)} = \sqrt{\mathrm{tr}(I_m)} = \sqrt{m}$ for $U \in U(m)$. Thus $d_{\mathrm{HS}}(\boldsymbol{\Phi}_{\mathbf{X}}(M), \boldsymbol{\Phi}_{\mathbf{Y}}(M)) \leq \sqrt{2}m$. We take expectation over $\mathbb{P}_{\mathcal{M}}$ to conclude. $\qquad\square$

The result below is formulated in terms of the Hilbert–Schmidt norm of matrices in $\mathbb{C}^{m\times m}$. Any other norm on $\mathbb{C}^{m\times m}$ is equivalent to that, modulo a constant depending on $m$ only. In fact, the strict inequality $\|T\|_{\mathrm{op}} \le \|T\|_{\mathrm{HS}}$ for $T \in \mathbb{C}^{m\times m}$ holds. See, *e.g.*, [48, Lemma 3.1.10, p.55].

**Lemma B.4.** *Let $L, \tilde{L} : [a,b] \to \mathbb{R}^d$ be two linear paths, and let $M \in \mathfrak{u}(m)^d := \mathcal{L}\left(\mathbb{R}^d, \mathfrak{u}(m)\right)$ as before. Denote by $\|\bullet\|_{\mathrm{e}}$ the usual Euclidean norm on $\mathbb{R}^d$ and $\||M\||$ the operator norm of $M : \left(\mathbb{R}^d, \|\bullet\|_{\mathrm{e}}\right) \to (\mathfrak{u}(m), \|\bullet\|_{\mathrm{HS}})$. Then we have*

$$\left\| e^{M(L(t))} - e^{M(\tilde{L}(t))} \right\|_{\mathrm{HS}} \le \||M\|| \left\| L(t) - \tilde{L}(t) \right\|_{\mathrm{e}} \qquad \text{for each } t \in [a,b].$$

*Proof.* Let $\Gamma(t,s) := M\left((1-s)L(t) + s\tilde{L}(t)\right)$ with $t \in [a,b]$ and $s \in [0,1]$. This is the linear interpolation between $\Gamma(t,0) = M \circ L(t)$ and $\Gamma(t,1) = M \circ \tilde{L}(t)$. Then we have

$$\left\| e^{ML(t)} - e^{M\tilde{L}(t)} \right\|_{\mathrm{HS}} = \left\| \int_0^1 \frac{\partial e^{\Gamma}}{\partial s}(t,s)\,\mathrm{d}s \right\|_{\mathrm{HS}}$$
$$= \left\| \int_0^1 \int_0^1 e^{(1-r)\Gamma(t,1-s)} \frac{\partial \Gamma}{\partial s}(t,s) e^{r\Gamma(t,s)}\,\mathrm{d}r\,\mathrm{d}s \right\|_{\mathrm{HS}},$$

thanks to an identity for differentiation of matrix exponential and the inequality $\|T_1 T_2\|_{\mathrm{HS}} \le \|T_1\|_{\mathrm{op}} \|T_2\|_{\mathrm{HS}}$. Here $e^{(1-r)\Gamma(t,1-s)}$ and $e^{r\Gamma(t,s)}$ take values in $U(m)$, hence of operator norm 1 for any parameters $t, s, r$. So we infer that

$$\left\| e^{ML(t)} - e^{M\tilde{L}(t)} \right\|_{\mathrm{HS}} \le \int_0^1 \int_0^1 \left\| \frac{\partial \Gamma}{\partial s}(t,s) \right\|_{\mathrm{HS}}\,\mathrm{d}r\,\mathrm{d}s$$
$$= \left\| M\left(\tilde{L}(t) - L(t)\right) \right\|_{\mathrm{HS}} \le \||M\|| \left\| \tilde{L}(t) - L(t) \right\|_{\mathrm{e}},$$

where the first inequality holds for Bochner integrals. See [46, Corollary 1, p.133]. $\qquad \square$

**Lemma B.5** (Subadditivity of unitary feature). *Let $X, Y \in \mathrm{BV}\left([0,T]; \mathbb{R}^d\right)$ be BV-paths, and $\mathcal{U}_M$ be the unitary feature associated with $M \in \mathcal{L}\left(\mathbb{R}^d, \mathfrak{u}(m)\right) = \mathfrak{u}(m)^d$. For any $0 < t < T$ we have*

$$\|\mathcal{U}_M(X) - \mathcal{U}_M(Y)\|_{\mathrm{HS}} \le \|\mathcal{U}_M(X_{0,t}) - \mathcal{U}_M(Y_{0,t})\|_{\mathrm{HS}} + \|\mathcal{U}_M(X_{t,T}) - \mathcal{U}_M(Y_{t,T})\|_{\mathrm{HS}}.$$

*Proof.* We apply the multiplicative property of unitary feature in Lemma A.5, the triangle inequality, and the unitary invariance of the Hilbert–Schmidt norm to estimate that

$$\|\mathcal{U}_M(X) - \mathcal{U}_M(Y)\|_{\mathrm{HS}}$$
$$= \|\mathcal{U}_M(X_{0,t}) \cdot \mathcal{U}_M(X_{t,T}) - \mathcal{U}_M(Y_{0,t}) \cdot \mathcal{U}_M(Y_{t,T})\|_{\mathrm{HS}}$$
$$\le \|(\mathcal{U}_M(X_{0,t}) - \mathcal{U}_M(Y_{0,t})) \cdot \mathcal{U}_M(X_{t,T})\|_{\mathrm{HS}} + \|\mathcal{U}_M(Y_{0,t})(\mathcal{U}_M(X_{t,T}) - \mathcal{U}_M(Y_{t,T}))\|_{\mathrm{HS}}$$
$$= \|\mathcal{U}_M(X_{0,t}) - \mathcal{U}_M(Y_{0,t})\|_{\mathrm{HS}} + \|\mathcal{U}_M(X_{t,T}) - \mathcal{U}_M(Y_{t,T})\|_{\mathrm{HS}}.$$

$\qquad \square$

**Proposition B.6.** *For $\mathbf{X}, \mathbf{Y} \in \mathcal{X}$, the unitary feature $\mathcal{U}_M(\mathbf{X})$ with $M \in \mathcal{L}\left(\mathbb{R}^d, \mathfrak{u}(m)\right) = \mathfrak{u}(m)^d$ satisfies*

$$\|\mathcal{U}_M(\mathbf{X}) - \mathcal{U}_M(\mathbf{Y})\|_{\mathrm{HS}} \le \||M\|| \, \mathrm{Tot.Var.}[\mathbf{X} - \mathbf{Y}],$$

*where $\mathrm{Tot.Var.}[\mathbf{X} - \mathbf{Y}]$ denotes the total variation over $[0,T]$ of the path $\mathbf{X} - \mathbf{Y}$.*

*Proof.* Given BV-paths $\mathbf{X}, \mathbf{Y}$ with the same initial point, there are piecewise linear approximations $\{\mathbf{X}^n\}, \{\mathbf{Y}^n\}$ with common partition $0 = t_0 < t_1 < \cdots < t_n = T$ converging respectively to $\mathbf{X}$ and $\mathbf{Y}$ in the $p$-variation metric; $p \in (1,2)$. Applying Lemma B.5 recursively, we obtain that

$$\|\mathcal{U}_M(\mathbf{X}^n) - \mathcal{U}_M(\mathbf{Y}^n)\|_{\mathrm{HS}} \le \sum_{i=0}^{n-1} \left\| \mathcal{U}_M(\mathbf{X}^n_{t_i, t_{i+1}}) - \mathcal{U}_M(\mathbf{Y}^n_{t_i, t_{i+1}}) \right\|_{\mathrm{HS}}$$

By definition of unitary feature and Lemma B.4, one deduces that

$$\|\mathcal{U}_M(\mathbf{X}^n) - \mathcal{U}_M(\mathbf{Y}^n)\|_{\mathrm{HS}} \leq \sum_{i=0}^{n-1} \||M\|| \left\| \mathbf{X}_{t_i, t_{i+1}}^n - \mathbf{Y}_{t_i, t_{i+1}}^n \right\|_{\mathrm{e}}.$$

We may now conclude by taking supremum over all partitions and sending $n \to \infty$ with the limit. This is a consequence of continuity of the Itô map with respect to the driving path in $p$-variation topology, since the vector field in (4) is Lipschitz ([27, Theorem 1.20]). $\square$

**Theorem B.7** (Dependence on continuous parameter). *Let $\mathcal{X}$ and $\mathcal{Z}$ be subsets of $\mathrm{BV}\left([0,T]; \mathbb{R}^d\right)$, $(\Theta, \rho)$ be a metric space, $\mathbb{Q}$ be a Borel probability measure on $\mathcal{Z}$, and $\mathcal{M}$ be a Borel probability measure on $\mathfrak{u}(m)^d$. Assume that $g : \Theta \times \mathcal{Z} \to \mathcal{X}$, $(\theta, \mathbf{Z}) \mapsto g_\theta(\mathbf{Z})$ is Lipschitz in $\theta$ such that $\mathrm{Tot.Var.}\left[g_\theta(\mathbf{Z}) - g_{\theta'}(\mathbf{Z})\right] \leq \omega(\mathbf{Z})\rho(\theta, \theta')$. In addition, suppose that $\mathbb{E}_{M \sim \mathbb{P}_\mathcal{M}}\left[\||M\||^2\right] < \infty$ and $\mathbb{E}_{\mathbf{Z} \sim \mathbb{Q}}\left[\omega(\mathbf{Z})\right] < \infty$. Then $\mathrm{PCFD}_\mathcal{M}\left(g_\theta(\mathbf{Z}), \mathbf{X}\right)$ is Lipschitz in $\theta$.*

*Moreover, it holds that*

$$\left|\mathrm{PCFD}_\mathcal{M}\left(g_\theta(\mathbf{Z}), \mathbf{X}\right) - \mathrm{PCFD}_\mathcal{M}\left(g_{\theta'}(\mathbf{Z}), \mathbf{X}\right)\right| \leq \sqrt{\mathbb{E}_{M \sim \mathbb{P}_\mathcal{M}}\left[\||M\||^2\right]}\, \mathbb{E}_{\mathbf{Z} \sim \mathbb{Q}}\left[\omega(\mathbf{Z})\right]\, \rho(\theta, \theta')$$

*for any $\theta, \theta' \in \Theta$, $\mathbf{Z} \in \mathcal{Z}$, $\mathbf{X} \in \mathcal{X}$, and $\mathcal{M} \in \mathcal{P}\left(\mathfrak{u}(m)^d\right)$.*

*Proof.* As $\mathrm{PCFD}_\mathcal{M}$ is a pseudometric (Lemma B.2), we have

$$\left|\mathrm{PCFD}_\mathcal{M}\left(g_\theta(\mathbf{Z}), \mathbf{X}\right) - \mathrm{PCFD}_\mathcal{M}\left(g_{\theta'}(\mathbf{Z}), \mathbf{X}\right)\right| \leq \mathrm{PCFD}_\mathcal{M}\left(g_\theta(\mathbf{Z}), g_{\theta'}(\mathbf{Z})\right).$$

We may control the right-hand side as follows, using subsequentially the definitions of PCFD and PCF, Proposition B.6, and the assumptions in this theorem:

$$\mathrm{PCFD}_\mathcal{M}\left(g_\theta(\mathbf{Z}), g_{\theta'}(\mathbf{Z})\right)$$

$$= \left\{ \int_{\mathfrak{u}(m)^d} \left\| \boldsymbol{\Phi}_{g_\theta(\mathbf{Z})}(M) - \boldsymbol{\Phi}_{g_{\theta'}(\mathbf{Z})}(M) \right\|_{\mathrm{HS}}^2 \, \mathrm{d}\mathbb{P}_\mathcal{M} \right\}^{\frac{1}{2}}$$

$$= \left\{ \int_{\mathfrak{u}(m)^d} \left\| \int_{\mathcal{Z}} \left[\mathcal{U}_M\left(g_\theta(\mathbf{Z})\right) - \mathcal{U}_M\left(g_{\theta'}(\mathbf{Z})\right)\right] \, \mathrm{d}\mathbb{Q}(\mathbf{Z}) \right\|_{\mathrm{HS}}^2 \, \mathrm{d}\mathbb{P}_\mathcal{M} \right\}^{\frac{1}{2}}$$

$$\leq \left\{ \int_{\mathfrak{u}(m)^d} \||M\||^2 \left\{ \int_{\mathcal{Z}} \mathrm{Tot.Var.}\left[g_\theta(\mathbf{Z}) - g_{\theta'}(\mathbf{Z})\right] \, \mathrm{d}\mathbb{Q}(\mathbf{Z}) \right\}^2 \, \mathrm{d}\mathbb{P}_\mathcal{M} \right\}^{\frac{1}{2}}$$

$$\leq \sqrt{\mathbb{E}_{M \sim \mathbb{P}_\mathcal{M}}\left[\||M\||^2\right]} \left\{ \int_{\mathcal{Z}} \omega(\mathbf{Z})\rho(\theta, \theta') \, \mathrm{d}\mathbb{Q}(\mathbf{Z}) \right\}.$$

This completes the proof. $\square$

The unitary feature is *universal* in the spirit of the Stone–Weierstrass theorem; *i.e.*, continuous functions on paths can be uniformly approximated by linear functionals on unitary features.

As $\widetilde{\mathrm{PCFD}}$ metrises the weak topology on the space of path-valued random variables, it emerges as a more sensible distance metric for training time series generations than metrics without this property; *e.g.*, the Jensen–Shannon divergence.

**Theorem B.8** (Metrisation of weak-star topology). *Let $\mathcal{K} \subset \mathcal{X}$ be a compact subset. Suppose that $\{\mathcal{M}_j\}_{j \in \mathbb{N}}$ is a countable dense subset in $\mathcal{P}\left(\mathcal{L}\left(\mathbb{R}^d, \bigoplus_{m \in \mathbb{N}} \mathfrak{u}(m)\right)\right)$. Then $\widetilde{\mathrm{PCFD}}$ defined by Eqn. 13 metrises the weak-star topology on $\mathcal{P}(\mathcal{K})$. That is, $\widetilde{\mathrm{PCFD}}(\mathbf{X}_n, \mathbf{X}) \to 0 \iff \mathbf{X}_n \overset{d}{\to} \mathbf{X}$ as $n \to \infty$, where $\overset{d}{\to}$ denotes convergence in distribution of random variables.*

The metrisability of $\mathcal{P}(\mathcal{K})$ follows from general theorems in functional analysis: $\mathcal{K}$ is a compact metric space, hence $C^0(\mathcal{K})$ is separable ([13, Lemma 3.23]). Then, viewing $\mathcal{P}(\mathcal{K})$ as the unit circle in $\left[C^0(\mathcal{K})\right]^*$ via Riesz representation, we infer from [13, Proposition 3.24] that $\mathcal{P}(\mathcal{K})$ is metrisable in the weak-star topology, which is equivalent to the distributional convergence of random variables.

*Proof.* The backward direction is straightforward. By the Riesz representation theorem of Radon measures, the distributional convergence is equivalent to that $\int_{\mathcal{K}} f \, d\mathbb{P}_{\mathbf{X}_n} \to \int_{\mathcal{K}} f \, d\mathbb{P}_{\mathbf{X}}$ for all continuous $f \in C(\mathcal{K})$. Thus $\left\| \int_{\mathcal{K}} \mathcal{U}_M \, d\mathbb{P}_{\mathbf{X}_n} - \int_{\mathcal{K}} \mathcal{U}_M \, d\mathbb{P}_{\mathbf{X}} \right\|_{HS} \to 0$, namely that $\mathbf{\Phi}_{\mathbf{X}_n}[M] \to \mathbf{\Phi}_{\mathbf{X}}[M]$ for each $M \in \mathcal{L}\left(\mathbb{R}^d; \mathfrak{u}(m)\right)$. The unitary feature $\mathcal{U}_M$ is bounded as it is $U(m)$-valued for some $m$, so we deduce from the dominated convergence theorem that $\widetilde{\mathrm{PCFD}}(\mathbf{X}_n, \mathbf{X}) \to 0$.

Conversely, suppose that $\widetilde{\mathrm{PCFD}}(\mathbf{X}_n, \mathbf{X}) \to 0$. Then

$$\int_{\mathcal{L}(\mathbb{R}^d; \mathfrak{u}(m))} \left\| \int_{\mathcal{K}} \mathcal{U}_M \, d\mathbb{P}_{\mathbf{X}_n} - \int_{\mathcal{K}} \mathcal{U}_M \, d\mathbb{P}_{\mathbf{X}} \right\|_{HS}^2 \, d\mathbb{P}_{\mathcal{M}}(M) = 0$$

for any $m \in \mathbb{N}$ and $\mathcal{M} \in \mathcal{P}\left(\mathcal{L}\left(\mathbb{R}^d; \mathfrak{u}(m)\right)\right)$, in particular for those with full support. In view of the universality Theorem A.8 proved above, for any fixed $\epsilon > 0$ and any continuous function $f \in C^0(\mathcal{K})$, by approximating $f$ with sum of finitely many $L_i \circ \mathcal{U}_{M_i}$ (the notations are as in Theorem A.8), one infers that for $n$ and $m_\star$ sufficiently large, it holds that

$$\int_{\mathcal{L}(\mathbb{R}^d; \mathfrak{u}(m_\star))} \left| \int_{\mathcal{K}} f \, d\mathbb{P}_{\mathbf{X}_n} - \int_{\mathcal{K}} f \, d\mathbb{P}_{\mathbf{X}} \right|^2 \, d\mathbb{P}_{\mathcal{M}}(M) < \epsilon.$$

By considering those measures with $\mathrm{spt}(M) = \mathcal{L}\left(\mathbb{R}^d; \mathfrak{u}(m_\star)\right)$, we deduce that

$$\lim_{n \to \infty} \left| \int_{\mathcal{K}} f \, d\mathbb{P}_{\mathbf{X}_n} - \int_{\mathcal{K}} f \, d\mathbb{P}_{\mathbf{X}} \right| = 0 \qquad \text{for any } f \in C^0(\mathcal{K}).$$

This is tantamount to the distributional convergence. $\qquad \square$

*Proof.* We first prove the 'if' direction of the statement. By the Portmanteau theorem [21], convergence in distribution $X_n \xrightarrow{d} X$ implies, for any bounded continuous map $f$, we have $\mathbb{E}_{x \sim \mathbb{P}_n}[f(x)] \to \mathbb{E}_{x \sim \mathbb{P}}[f(x)]$. Therefore, for any $M \in \mathcal{L}(\mathbb{R}^d, \mathfrak{u}(m))$, $\mathbb{E}_{x \sim \mathbb{P}_n}[\mathcal{U}_M(x)] \to \mathbb{E}_{x \sim \mathbb{P}}[\mathcal{U}_M(x)]$, which implies $\|\Phi_{X_n}(M) - \Phi_X(M)\|_{HS}^2 \to 0$ as $n \to \infty$. Hence, it follows that, as $n \to \infty$,

$$PCFD(X_n, X) := \mathbb{E}_{M \sim \mathbb{P}_{\mathcal{M}}} \|\Phi_{X_n}(M) - \Phi_X(M)\|_{HS}^2 \to 0,$$

which completes the proof of 'if' direction.

Now we proceed with the 'only if' direction. By the universality of the unitary path development from Theorem A.8, for any continuous function $f : \mathcal{K} \to \mathbb{C}$ and $\epsilon > 0$, there exist $M_1, \cdots, M_N \in \mathcal{L}(\mathbb{R}^d, \mathfrak{u}(m))$ and $L_1, \ldots, L_N \in \mathcal{L}(\mathcal{U}(m); \mathbb{C})$ such that

$$\left| \mathbb{E}_{x \sim \mathbb{P}}[f(x)] - \sum_{i=1}^{N} L_i \circ \mathbb{E}_{x \sim \mathbb{P}}[\mathcal{U}_{M_i}(x)] \right| < \epsilon. \tag{14}$$

or equivalently $\left| \mathbb{E}_{x \sim \mathbb{P}}[f(x)] - \sum_{i=1}^{N} L_i \circ \Phi_X(M_i) \right| < \epsilon$. For simplicity, we denote $\mathbb{E}_{x \sim \mathbb{P}_n}$ and $\mathbb{E}_{x \sim \mathbb{P}}$ as $\mathbb{E}_n$ and $\mathbb{E}$ respectively. Therefore,

$$|\mathbb{E}_n[f(x)] - \mathbb{E}[f(x)]| \leq \left| \mathbb{E}_n[f(x)] - \sum_{i=1}^{N} L_i \circ \Phi_{X_n}(M_i) \right| + \left| \mathbb{E}[f(x)] - \sum_{i=1}^{N} L_i \circ \Phi_X(M_i) \right| \tag{15}$$

$$+ \left| \sum_{i=1}^{N} L_i \circ (\Phi_{X_n}(M_i) - \Phi_X(M_i)) \right| \tag{16}$$

$$\leq 2\epsilon + \sum_{i=1}^{N} |L_i|_{op} \|\Phi_{X_n}(M_i) - \Phi_X(M_i)\|_{HS}^2 \tag{17}$$

where $|L|_{op} := \sup_{x \in \mathcal{U}(m) \setminus 0} \frac{|L(x)|}{\|x\|_{HS}^2}$ the operator norm. Since $PCFD(X_n, X) := \mathbb{E}_{M \sim \mathbb{P}_{\mathcal{M}}} \|\Phi_{X_n}(M) - \Phi_X(M)\|_{HS}^2 \to 0$ as $n \to \infty$ and $\epsilon$ is arbitrary, $\mathbb{E}_{x \sim \mathbb{P}_n}[f(x)] \to \mathbb{E}_{x \sim \mathbb{P}}[f(x)]$ for any continuous bounded function $f : \mathcal{K} \to \mathbb{C}$, which implies $X_n \xrightarrow{d} X$ by the Portmanteau theorem [21]. $\qquad \square$

## B.3 Relation with MMD

We now discuss linkages between PCFD and MMD (maximum mean discrepancy) defined over $\mathcal{P}(\mathcal{X})$, the space of Borel probability measures (equivalently, probability distributions) on $\mathcal{X}$.

**Definition B.9.** Given a kernel function $\kappa : \mathcal{X} \times \mathcal{X} \to \mathbb{R}$, the MMD associated to $\kappa$ is the function $\mathrm{MMD}_\kappa : \mathcal{P}(\mathcal{X}) \times \mathcal{P}(\mathcal{X}) \to \mathbb{R}^+$ given as follows: for independent random variables $\mathbf{X}, \mathbf{Y}$ on $\mathcal{X}$, set

$$\mathrm{MMD}_\kappa^2(\mathbb{P}_\mathbf{X}, \mathbb{P}_\mathbf{Y}) = \mathbb{E}_{\mathbf{X}, \mathbf{X}' \overset{\mathrm{iid}}{\sim} \mathbb{P}_\mathbf{X}}[\kappa(\mathbf{X}, \mathbf{X}')] + \mathbb{E}_{\mathbf{Y}, \mathbf{Y}' \overset{\mathrm{iid}}{\sim} \mathbb{P}_\mathbf{Y}}[\kappa(\mathbf{Y}, \mathbf{Y}')] - 2\mathbb{E}_{\mathbf{X} \sim \mathbb{P}_\mathbf{X}, \mathbf{Y} \sim \mathbb{P}_\mathbf{Y}}[\kappa(\mathbf{X}, \mathbf{Y})].$$

The PCFD can be interpreted as an MMD on measures of the path space with a specific kernel. Compare with [40] for the case of $\mathbb{R}^d$.

**Proposition B.10** (PCFD as MMD)**.** *Given $\mathcal{M} \in \mathcal{P}(\mathfrak{u}(m)^d)$ and $\mathcal{X}$-valued random variables $\mathbf{X}$ and $\mathbf{Y}$ with induced distributions $\mathbb{P}_\mathbf{X}$ and $\mathbb{P}_\mathbf{Y}$, resp. Then $\mathrm{PCFD}_\mathcal{M}(\mathbf{X}, \mathbf{Y}) = \mathrm{MMD}_\kappa(\mathbb{P}_\mathbf{X}, \mathbb{P}_\mathbf{Y})$ with kernel $\kappa(\mathbf{x}, \mathbf{y}) := \mathbb{E}_{M \sim \mathbb{P}_\mathcal{M}}\left[\left\|\mathcal{U}_M(\mathbf{x}) - \mathcal{U}_M(\mathbf{y})\right\|_{\mathrm{HS}}\right] = \mathbb{E}_{M \sim \mathbb{P}_\mathcal{M}}\left[\mathrm{tr}\left(\mathcal{U}_M\left(\mathbf{x} \star \overleftarrow{\mathbf{y}}\right)\right)\right]$.*

Throughout, $\star$ designates concatenation of paths and $\overleftarrow{\mathbf{y}}$ is the path obtained by running $\mathbf{y}$ backwards. The operation $\mathbf{x} \star \overleftarrow{\mathbf{y}}$ on the path space is analogous to $\mathbf{x} - \mathbf{y}$ on $\mathbb{R}^d$. If $\mathbf{y} = \mathbf{x}$, then $\mathbf{x} \star \overleftarrow{\mathbf{y}}$ is the null path. See the Appendix for proofs and further discussions.

*Remark* B.11 (Computational cost complexity). By Proposition B.10, PCFD is an MMD. However, to compute EPCFD, we may directly calculate the expected distance between the PCFs, without going over the kernel calculations in the MMD approach. Our method is significantly more efficient, especially for large datasets. The computational complexity of EPCFD is linear in sample size, whereas the MMD approach is quadratic.

*Proof.* By definition of PCFD, we have

$$\mathrm{PCFD}_\mathcal{M}^2(\mu, \nu) = \mathbb{E}_{M \sim \mathbb{P}_\mathcal{M}}\left[\|\Phi_\mathbf{X}(M) - \Phi_\mathbf{Y}(M)\|_{\mathrm{HS}}^2\right]$$
$$= \mathbb{E}_{M \sim \mathbb{P}_\mathcal{M}}\left[\|\Phi_\mathbf{X}(M)\|_{\mathrm{HS}}^2 + \|\Phi_\mathbf{Y}(M)\|_{\mathrm{HS}}^2 - 2\langle\Phi_\mathbf{X}(M), \Phi_\mathbf{Y}(M)\rangle_{\mathrm{HS}}\right]$$

where $\Phi_\mathbf{X}(M) = \mathbb{E}_{\mathbf{X} \sim \mu}[\mathcal{U}_M(\mathbf{X})]$ and $\Phi_\mathbf{Y}(M) = \mathbb{E}_{\mathbf{Y} \sim \mu}[\mathcal{U}_M(\mathbf{Y})]$, respectively. Using Fubini's theorem and observing that $\langle\Phi_\mathbf{X}(M), \Phi_\mathbf{Y}(M)\rangle_{\mathrm{HS}} \in L^2(\mathbb{P}_\mathcal{M})$ (as $\Phi_\mathbf{X}(M)$ and $\Phi_\mathbf{Y}(M)$ are $U(m)$-valued, they indeed lie in $L^\infty(\mathbb{C}^{m \times m}; \mathbb{P}_\mathcal{M})$ as $U(m)$ is a compact Lie group under the Hilbert–Schmidt metric), we deduce that

$$\mathbb{E}_{M \sim \mathbb{P}_\mathcal{M}}[\langle\Phi_\mathbf{X}(M), \Phi_\mathbf{Y}(M)\rangle_{\mathrm{HS}}] = \mathbb{E}_{\mathbf{X} \sim \mu}[\mathbb{E}_{\mathbf{Y} \sim \nu}[\mathbb{E}_{M \sim \mathbb{P}_\mathcal{M}}[\langle\mathcal{U}_M(\mathbf{X}), \mathcal{U}_M(\mathbf{Y})\rangle_{\mathrm{HS}}]]].$$

The first equality then follows from the identification $\kappa(\mathbf{x}, \mathbf{y}) = \mathbb{E}_{M \sim \mathbb{P}_\mathcal{M}}[\langle\mathcal{U}_M(\mathbf{x}), \mathcal{U}_M(\mathbf{y})\rangle_{\mathrm{HS}}]$ and the definition of $\mathrm{MMD}_\kappa$.

On the other hand, by Lemma A.5 and the definition of the Hilbert–Schmidt inner product on $U(m)$, one may rewrite the kernel function as follows:

$$\kappa(\mathbf{x}, \mathbf{y}) = \mathbb{E}_{M \sim \mathbb{P}_\mathcal{M}}\left[\langle\mathcal{U}_M(\mathbf{x}), \mathcal{U}_M(\mathbf{y})\rangle_{\mathrm{HS}}\right]$$
$$= \mathbb{E}_{M \sim \mathbb{P}_\mathcal{M}}\left[\mathrm{tr}(\mathcal{U}_M(\mathbf{x}) \cdot \mathcal{U}_M^{-1}(\mathbf{y}))\right] = \mathbb{E}_{M \sim \mathbb{P}_\mathcal{M}}\left[\mathrm{tr}\left(\mathcal{U}_M\left(\mathbf{x} \star \overleftarrow{\mathbf{y}}\right)\right)\right],$$

where $\star$ denotes the concatenation of paths. The second equality now follows. $\qquad\square$

## B.4 Empirical PCFD

### B.4.1 Initialisation of $\mathcal{M}$

A linear map $M \in \mathcal{L}(\mathbb{R}^d, \mathfrak{u}(m))$ can be canonically represented by $d$ independent anti-Hermitian matrices $M_1, \ldots, M_d \subset \mathfrak{u}(m) \in \mathbb{C}^{m \times m}$. To sample empiracal distribution of $\mathcal{M} \in \mathcal{P}[\mathcal{L}(\mathbb{R}^d, \mathfrak{u}(m))]$ from $\mathbb{P}_\mathcal{M}$, we propose a sampling scheme over $\mathfrak{u}(m)$. This can also be used as an effective initialisation of model parameters $\theta_M \in \mathfrak{u}(m)^{d \times k}$ for the empirical measure of $\mathcal{M}$.

In practice, when working with the Lie algebra $\mathfrak{u}(m)$, *i.e.*, the vector space of $m \times m$ complex-valued matrices that are anti-Hermitian ($A^* + A = 0$, where $A^*$ is the transpose conjugate of $A$), we view each anti-Hermitian matrix as an $2m \times 2m$ *real* matrix via the isomorphism of $\mathbb{R}$-vector spaces $\mathbb{R}^{2m \times 2m} \cong \mathbb{C}^{m \times m}$.

Under the above identification, we have the decomposition

$$\mathfrak{u}(m) \cong \mathfrak{o}(m) \oplus \sqrt{-1} \left( \mathrm{Sym}_{m \times m}/\mathfrak{z}(m) \right) \oplus \sqrt{-1}\mathfrak{z}(m), \tag{18}$$

where $\mathfrak{o}(m)$ is the Lie algebra of anti-symmetric $m \times m$ real matrices, $\mathrm{Sym}_{m \times m}$ is the space of $m \times m$ real symmetric matrices, $\mathfrak{z}(m)$ consists of $m \times m$ real diagonal matrices and $\mathrm{Sym}_{m \times m}/\mathfrak{z}(m)$ denotes the quotient space of real symmetric matrices by the real diagonal matrices.

The sampling procedure of $\mathbb{P}_{\mathcal{M}}$, is given as follows. First, we simulate $\mathbb{R}^{m \times m}$ valued and i.i.d random variables $A$ and $B$, whose elements are i.i.d and satisfy the pre-specified distribution in $\mathcal{P}(\mathbb{R})$. We have the decomposition $B = D \oplus E$, where D and E are a diagonal random matrix and a off-diagonal random matrix respectively. Then we construct the anti-symmetric matrix $R = \frac{1}{\sqrt{2}}(A^T - A)$ and matrix in the quotient space $\mathrm{Sym}_{m \times m}/\mathfrak{z}(m)$, $C = \frac{1}{\sqrt{2}}(E^T + E)$, and diagonal matrix $D$. Correspondingly, we simulate $\mathfrak{u}(m)$-valued random variables by virtue of Eq. (18). As the empirical measure of the $\mathcal{M}$ can be fully characterised by the model parameters $\theta_M \in \mathfrak{u}(m)^{d \times k}$, we sample $d \times k$ i.i.d. samples which take values in $u(m)$.

### B.4.2 Hypothesis test

In the following, we illustrate the efficacy of the proposed *trainable EPCFD* metric in the context of the hypothesis test on stochastic processes.

**Example B.12** (Hypothesis testing on fractional Brownian motion). *Consider the 3-dimensional Brownian motion* $\mathbf{B} := (B_t)_{t \in [0,T]}$ *and the fraction Brownian motion* $\mathbf{B}^h := (B_t^h)_{t \in [0,T]}$ *with the Hurst parameter h. We simulated 5000 sample paths for both* $\mathbf{B}$ *and* $\mathbf{B^h}$ *with 50 discretized time steps. We apply the proposed optimized EPCFD metric to the two-sample testing problem: the null hypothesis* $H_0 : \mathbf{B} \stackrel{d}{=} \mathbf{B}^h$ *against the alternative* $H_1 : \mathbf{B} \stackrel{d}{\neq} \mathbf{B}^h$. *We compare the optimized EPCFD metric with EPCFD metric with the prespecified distribution (PCF) and the characteristic function distance (CF) on the flattened time series [25]. The optimized PCFs are trained on a separate set of 5000 training samples to maximise the PCFD. The details of training can be found at Appendix C.2.*

*We conduct the permutation test to compute the power of a test (i.e. the probability of correct rejection of the null* $H_0$*) and Type I error (i.e. the probability of false acceptances of the null* $H_0$*) for varying* $h \in \{0.2 + 0.1 \cdot i\}_{i=0}^6$. *Note that when* $h = 0.5$, $\mathbf{B}$ *and* $\mathbf{B}^h$ *have the same distribution and hence are indistinguishable. Therefore, the better the test metric is, the test power should be closer to 0 when h is close to 0.5, whereas it should be closer to 1 when h is away from* $0.5$. *We refer to [22] for more in-depth information on hypothesis testing and permutation test statistics.*

*The plot of the test power and Type 1 error in Figure 6 shows that CF fails in the two sample tests, whilst both EPCFD and optimised EPCFD can distinguish the samples from the stochastic process when* $h \neq 0.5$. *It indicates that the EPCFD captures the distribution of time series much more effectively than the conventional CF metric. Moreover, optimization of EPCFD increases the test power while decreasing the type1 error, particularly when h is closer to* $0.5$.

## C  Numerical experiments

### C.1  Experimental detail on PCF-GAN

#### C.1.1  General notes

**Codes.** The code for reproducing all experiments can be found in `https://github.com/DeepIntoStreams/PCF-GAN`.

**Software.** We conducted all experiments using PyTorch 1.13.1 [34] and performed hyperparameter tuning with Wandb [5]. To ensure reproducibility, we implemented benchmark models based on open-source code from [45, 43, 12]. We used the Ksig library [42] to calculate the Sig-MMD metrics. The codes in [25] were used to compute characteristics function distance in Example B.12.

**Computing infrastructure.** The experiments were performed on a computational system running Ubuntu 22.04.2 LTS, comprising three Quadro RTX 8000 and two RTX A6000 GPUs. Each experiment was run independently on a single GPU, with the training phase taking between 6 hours to 3 days, depending on the dataset and models used.

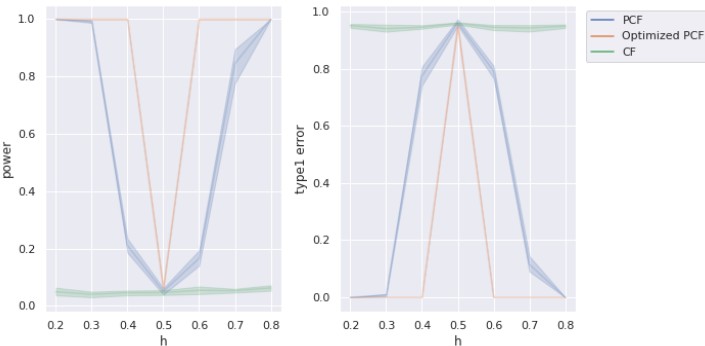

Figure 6: Plots of the test power (**Left**) and the Type-I error (**Right**) against the Hurst parameter $h \in [0.2, 0.8]$ on the two sample tests for the Brownian motion $\mathbf{B}$ against Fractional Brownian motions ($\mathbf{B}^h$) by using three metrics, i.e., *PCFD*, optimized *EPCFD* and *CFD*.

**Architectures.** To ensure a fair comparison, we employed identical network architectures, with two layers of LSTMs having 32 hidden units, for both the generator and discriminator across all models. For the generator, the output of the LSTM (full sequence) was passed through a Tanh activation function and a linear output layer. All generative models take a multi-dimensional discretized Brownian motion as the noise distribution, scaling it to ensure values were controlled within the range $[-1, 1]$. The dimension and scaling factor varied based on the dataset and were specified in the individual sections as below.

The PCF-GAN uses the development layers on the unitary matrix [26] to calculate the PCFD distance. For all experiments, we fixed the unitary matrix size and coefficient $\lambda_2$ for the regularization loss to 10 and 1, respectively. The number of unitary linear maps and the coefficient $\lambda_1$ of the recovery loss were determined via hyper-parameter tuning, which varied depending on the dataset (see individual section for details).

Regarding TimeGAN, the following approach described in [45] and employed embedding, supervisor, and recovery modules. Each of these modules had two layers of LSTMs with 32 hidden units. For COT-GAN, we used two separate modules for discriminators, each with two layers of LSTMs with 32 hidden units. Based on the recommendation from COT-GAN [43] and informal hyperparameter tuning, we set $\lambda = 10$ and $\epsilon = 1$ for all experiments.

**Optimisation & training.** We used the ADAM optimizer for all experiments [20], with a learning rate of 0.001 for both generators and discriminators. The learning rate for the unitary development network is 0.005. The initial decay rates in the ADAM optimizer are set $\beta_1 = 0$, $\beta_2 = 0.9$. The discriminator was trained for two iterations per iteration of the generator's training. For TimeGAN, we followed the training scheme for each module as suggested in the original paper. The batch size was 64 for all experiments. These hyperparameters do not substantially affect the results.

To improve the training stability of GAN, we employed three techniques. Firstly, we applied a constant exponential decay rate of 0.97 to the learning rate for every 500 generator training iterations. Secondly, we clipped the norm of gradients in both generator and discriminator to 10. Thirdly, we used the Cesaro mean of the generator weights after certain iterations to improve the performance of the final model, as suggested by [44]. In all cases, we selected the number of training iterations such that all methods could produce stable generative samples. The optimal number of training iterations and weight averaging scheme varied for each dataset. More details can be found in the respective sections.

**Test metrics.** *Discriminative score.* The network architecture of the post-hoc classifier consists of two layers of LSTMs with 16 hidden units. The dataset was split into equal proportions of real and generated time series with labels 0 and 1, with an $80\%$ / $20\%$ train/test split for training and evaluation. The discriminative model was trained for 30 epochs using Adam with a learning rate of 0.001 and a batch size of 64. The best classification error on the test set was reported.

*Predictive score.* The network architecture of the post-hoc sequence-to-sequence regressor consists of two layers of LSTMs with 16 hidden units. The model was trained on the generated time series

and evaluated on the real time series, using the first $80\%$ of the time series to predict the last $20\%$. The predictive model was trained for 50 epochs using Adam with a learning rate of 0.001 and a batch size of 64. The best mean squared error on the test set was reported.

*Sig-MMD.* We directly computed the Sig-MMD by taking inputs of the real time series samples and generated time series samples. We used the radial basis function kernel applying to the truncated signature feature up to depth 5.

## C.2 Time dependent Ornstein-Uhlenbeck process

On this dataset, we experimented with the basic version of PCF-GAN, which only utilized the EPCFD as the discriminator without the autoencoder structure. The batch size is 256. The model are trained with 20000 generator training iterations and weight averaging on the generator was performed over the final 4000 generator training iterations. We used the 2-dimensional discretized Brownian motion as the noise distribution.

### C.2.1 Rough volatility model

We followed [31] considering a rough stochastic volatility model for an asset price process $(S_t)_{t \in [0,1]}$, which satisfies the below stochastic differential equation,

$$dS_t = \sqrt{V_t} S_t dZ_t, \tag{19}$$

$$V_t := \xi(t) exp \left( \eta B_t^H - \frac{1}{2} \eta^2 t^{2H} \right), \tag{20}$$

where $\xi(t)$ denotes the forward variance and $B_t^H$ denotes the frational Brownian motion (fBM) given by

$$B_t^H := \int_0^t K(t-s) dB_s, \quad K(r) := \sqrt{2H} r^{H-0.5}$$

where $(Z_t)_{t \in [0,1]}, (B_t)_{t \in [0,1]}$ are (possibly correlated) Brownian motions. In our experiments, the synthetic dataset is sampled from Equation (19) with $t \in [0,1]$, $H = 0.25$, $\xi(t) \sim \mathcal{N}(0.1, 0.01)$, $\eta = 0.5$ and initial condition $\log(S_0) \sim \mathcal{N}(0, 0.05)$. Each sample path is sampled uniformly from $[0,1]$ with the time discretization $\delta t = 0.005$, which consists of 200 time steps. We train the generators to learn the joint distribution of the log price and log volatility.

All methods are trained with 30000 generator training iterations and weight averaging on the generator was performed over the final 5000 generator training iterations. The input noise vectors have 5 dimension and 200 time steps.

For PCF-GAN, the coefficient $\lambda_1$ for the recovery loss was 50, and the number of unitary linear maps was 6.

### C.2.2 Stocks

We selected 10 large market cap stocks, which are Google, Apple, Amazon, Tesla, Meta, Microsoft, Nvidia, JP Morgan, Visa and P&G, from 2013 to 2021. The dataset consists of 5 features, including daily open, close, high, low prices and volume, available on `https://finance.yahoo.com/lookup`. We truncated the long stock time series into 20 days. The data were normalized with standard Min-Max normalisation on each feature channel. The Stock dataset used in our study is similar to the one employed in [25] but with a broader range of assets. Unlike the previous approach, we avoided sampling the time series using rolling windows with a stride of 1 to mitigate the presence of strong dependencies between samples.

All methods are trained with 30000 generator training iterations and weight averaging on the generator was performed over the final 5000 generator training iterations. The input noise vectors have 5 feature dimensions and 20 time steps.

For PCF-GAN, the coefficient $\lambda_1$ for the recovery loss was 400, and the number of unitary linear maps was 6.

### C.2.3 Beijing Air Quality

We used a dataset of the air quality in Beijing from the UCI repository [47] and available on `https://archive.ics.uci.edu/ml/datasets/Beijing+Multi-Site+Air-Quality+Data`. Each sample is a 10-dimensional time series of the SO2, NO2, CO, O3, PM2.5, PM10 concentrations, temperature, pressure, dew point temperature and wind speed. Each time series is recorded hourly over the course of a day. The data were normalized with standard Min-Max normalisation on each feature channel.

All methods are trained with 20000 generator training iterations and weight averaging on the generator was performed over the final 4000 generator training iterations. The input noise vectors have 5 dimensions and 24 time steps.

For PCF-GAN, the coefficient $\lambda_1$ for the recovery loss was 50, and the number of unitary linear maps was 6.

### C.2.4 EEG

We obtained the EEG eye state dataset from `https://archive.ics.uci.edu/ml/datasets/EEG+Eye+State`. The data is from one continuous EEG measurement on 14 variables with 14980 time steps. We truncated the long time series into smaller ones with 20 time steps. The data are subtracted by channel-wise mean, divided by three times the channel-wise standard deviation, and then passed through a tanh nonlinearity.

All methods are trained with 30000 generator training iterations and weight averaging on the generator was performed over the final 5000 generator training iterations. The input noise vectors have the 8 dimensional and 20 time steps.

For PCF-GAN, the coefficient $\lambda_1$ for the recovery loss was 50, and the number of unitary linear maps was 8.

## D  Supplementary results

### D.1  Ablation study

An ablation study was conducted on the PCF-GAN model to evaluate the importance of its various components. Specifically, the reconstruction loss and regularization loss were disabled in order to assess their impact on model performance across benchmark datasets and various test metrics. Table 3 consistently demonstrated that the PCF-GAN model outperformed the ablated versions, confirming the significance of these two losses in the overall model performance.

Table 3: Ablation study of PCF-GAN

| Dataset | Test Metrics | PCF-GAN | w/o $L_{\text{recovery}}$ | w/o $L_{\text{regularization}}$ | w/o $L_{\text{regularization}}$ & $L_{\text{recovery}}$ |
|---|---|---|---|---|---|
| RV | Discriminative | .0108±.006 | .0178±.017 | .0152±.020 | **.0101±.007** |
| | Predictive | .0390±.000 | **.0389±.000** | .0390±.003 | .0391±.001 |
| | Sig-MMD | **.0024±.001** | .0037±.001 | .0036 ±.002 | .0027±.001 |
| Stock | Discriminative | **.0784±.028** | .0963±.011 | .2538±.052 | .0815±.001 |
| | Predictive | .0125±.000 | **.0123±.000** | .0127±.000 | .0126±.001 |
| | Sig-MMD | **.0017±.000** | .0062±.002 | .0024±.001 | .0021±.001 |
| Air | Discriminative | **.2326±.058** | .3940±.068 | .4783±.029 | .3875±.009 |
| | Predictive | **.0237±.000** | .0239±.000 | .0283±.001 | .0240±.000 |
| | Sig-MMD | **.0126±.005** | .0111±.003 | .0232±.004 | .0163±.004 |
| EEG | Discriminative | **.3660±.025** | .4942±.010 | .5000±.000 | .4649±.015 |
| | Predictive | **.0246±.000** | .0299±.000 | .0636±.007 | .0248±.000 |
| | Sig-MMD | **.0180±.004** | .0296±.008 | 1.197±.234 | .0278±007 |

‘

Notably, the inclusion of the two additional losses significantly improved model performance on high-dimensional time series datasets, such as Air Quality and EEG, indicating that the proposed auto-encoder architecture effectively learns meaningful low-dimensional sequential embeddings.

Conversely, the exclusive use of the reconstruction loss led to a notable decrease in model performance, suggesting that the $l^2$ samplewise distance might not be suitable for time series data. However, the additional regularization loss helped overcome this issue by ensuring that the sequential embedding space is confined to a predetermined noise space, such as the discretized Brownian motion. As a result, the regularization loss helped to mitigate the problems that arose when relying solely on the reconstruction loss.

## D.2 Generated samples

In this section, we present random samples from the four benchmark datasets generated by PCF-GAN, TimeGAN, RGAN, and COT-GAN. Although interpreting the sample plots of the generated time series poses a challenge, our observations reveal that PCF-GAN successfully generates time series that capture the temporal dependencies exhibited in the original time series across all datasets. Conversely, COT-GAN generates trajectories that are relatively smoother compared to the real time series samples, demonstrated on Stock and EEG datasets, by Figure 8 and Figure 10 respectively. Figure 10 shows that TimeGAN occasionally produces samples with higher oscillations than those found in the real samples.

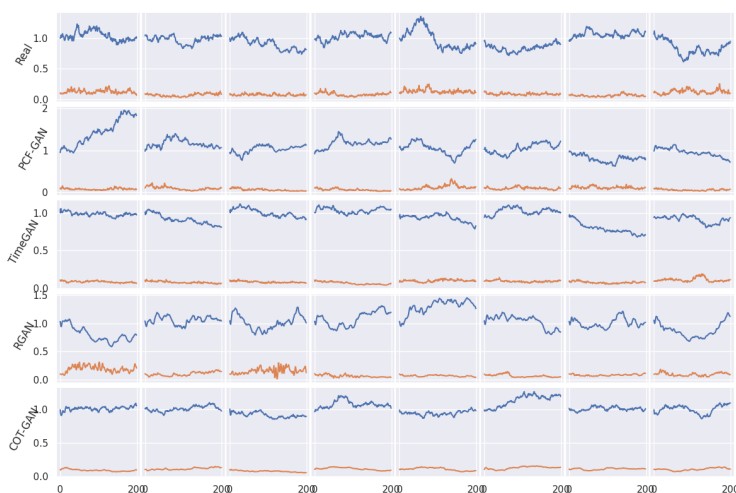

Figure 7: Generated samples from all models on Rough volatility dataset

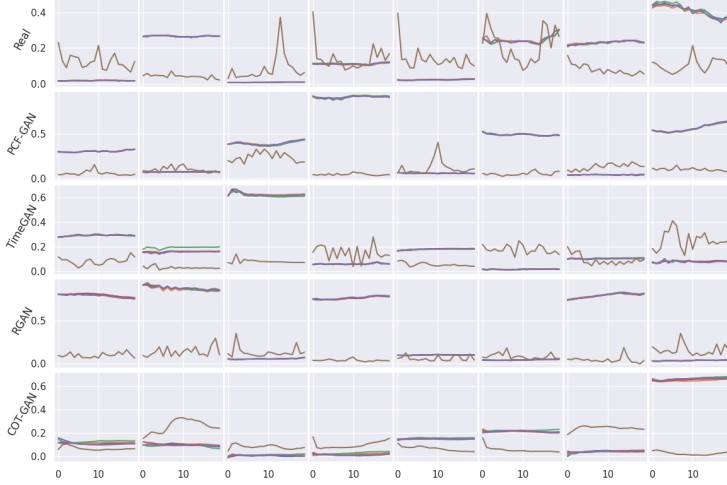

Figure 8: Generated samples from all models on Stock dataset

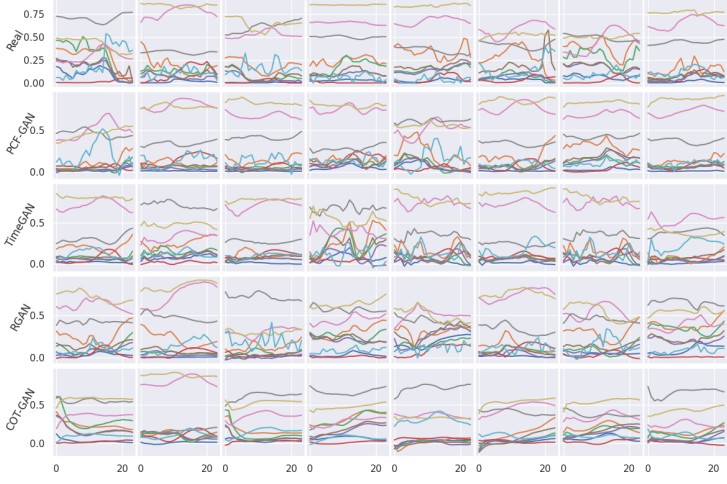

Figure 9: Generated samples from all models on Air Quality dataset

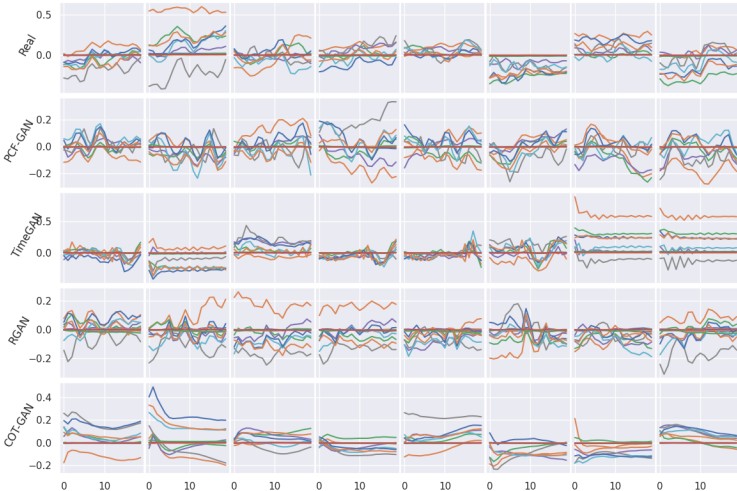

Figure 10: Generated samples from all models on EEG dataset

### D.3 Reconstructed samples

In this section, we present additional reconstructed time series samples generated by PCF-GAN and TimeGAN. Figure 11 illustrates that PCF-GAN consistently outperforms TimeGAN by producing higher-quality reconstructed samples across all datasets.

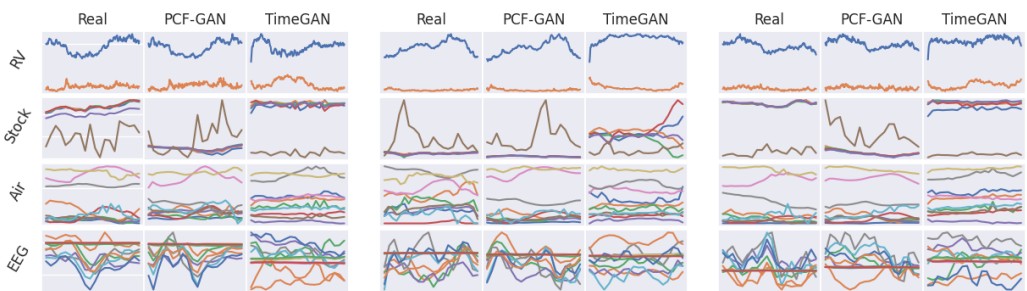

Figure 11: Reconstructed samples from PCF-GAN and TimeGAN on all benchmark datasets.

## D.4 Test metrics on (auto-)correlation and marginal distribution

This subsection details the supplementary test metrics in terms of fitting the autocorrelation, cross-correlation, and marginal distribution, as presented in Table 4. This table confirms that our proposed PCF-GAN consistently outperforms the benchmarking models across all datasets.

Table 4: Performance comparison of PCF-GAN and baselines on auto-correlation, cross-correlation and marginal distribution metrics. Best for each task is shown in bold.

| Task | | Generation | | | |
| --- | --- | --- | --- | --- | --- |
| *Dataset* | *Test Metrics* | *RGAN* | *COT-GAN* | *TimeGAN* | *PCF-GAN* |
| RV | *Auto-cor (lag 1)* | .0393±.001 | .0608±.001 | .0031±.001 | **.0022±.000** |
| | *Auto-cor (lag 5)* | .0134±.002 | .119±.002 | .0035±.002 | **.0030±.002** |
| | *Cross-cor (lag 0)* | .0193±.007 | .0234±.002 | **.0187±.011** | .0264±.011 |
| | *Cross-cor (lag 5)* | .0222±.007 | .1441±.012 | .0219±.010 | **.0158±.011** |
| | *Marginal Dist* | .311±1.13 | .2157±.306 | .1636±.223 | **.1234±.126** |
| Stock | *Auto-cor (lag 1)* | .127±.005 | .202±.0035 | .210±.005 | **.0123±.005** |
| | *Auto-cor (lag 5)* | .149±.009 | .267±.006 | .104±.006 | **.0187±.006** |
| | *Cross-cor (lag 0)* | .145±.031 | .169±.041 | .549±.034 | **.1815±.058** |
| | *Cross-cor (lag 5)* | .341±.031 | .456±.053 | .747±.038 | **.2510±.062** |
| | *Marginal Dist* | .3276±.044 | .2826±.061 | .4264±.063 | **.2730±.033** |
| Air | *Auto-cor (lag 1)* | .1678±.010 | .320±.006 | .1949±.006 | **.0927±.003** |
| | *Auto-cor (lag 5)* | .3226±.016 | .520±.028 | .5349±.034 | **.4739±.023** |
| | *Cross-cor (lag 0)* | 2.608±.106 | **1.942±.059** | 2.844±.0812 | 2.687±.149 |
| | *Cross-cor (lag 5)* | 3.181±.101 | 2.176±.116 | 2.536±.112 | **2.115±.121** |
| | *Marginal Dist* | .5527±.523 | .5142±.600 | .6229±.595 | **.5066±.572** |
| EEG | *Auto-cor (lag 1)* | 5.918±.116 | 6.202±.111 | 5.754±.083 | **5.668±.079** |
| | *Auto-cor (lag 5)* | **4.285±.074** | 5.911±.107 | 5.265±.083 | 4.467±.127 |
| | *Cross-cor (lag 0)* | 51.16±.508 | 24.12±.702 | 26.84±.638 | **22.27±.550** |
| | *Cross-cor (lag 5)* | 47.97±.354 | 31.31±.920 | 25.95±.466 | **19.43±.412** |
| | *Marginal Dist* | 15.18±21.94 | **8.518±13.6** | 13.35±21.7 | 10.09±16.6 |

