# OpenReview forum: "PCF-GAN: generating sequential data via the characteristic function of measures on the path space"
_NeurIPS.cc/2023/Conference — NeurIPS 2023 poster_

### Official Review · Reviewer_9Utd · 2023-07-09

**Soundness:** 2 fair
**Presentation:** 2 fair
**Contribution:** 2 fair
**Rating:** 4
**Confidence:** 3

**Summary:**

This paper proposes  PCF-GAN, which aims to improve the effectiveness of the discriminator in differentiating the time series distributions by utilizing the path characteristic function (PCF) as a principled representation of the time series distribution within the discriminator. The authors also give the theoretical foundation of the PCF distance, proving its properties that ensure stable and feasible training of PCF-GAN. Extensive experiments show the effectiveness of the proposed method.

**Strengths:**

The newly proposed metric in the discriminator for handling sequential data is interesting. Also, the author provides theoretical evidence for the analytical properties of the proposed loss metric.


**Weaknesses:**

- The presentation of this work is kind of confusing, making it difficult to fully comprehend the content. Specifically, in Figure 3, it is unclear what the generator loss represents if it is computed on the embedding vector. Additionally, it is unclear why the generator and regularization loss can enhance the discriminative power in Equation 10. Clarifying these help audience better understand.
- The related work section consists of only one paragraph, which may overlook some relevant research in the field.
- The ablation study of every component is required to demonstrate the effectiveness individually.
- Regarding the sequential data, I am wondering how does this method work well on more challenging domains like video data since videos are also sequential in nature.




**Questions:**

Please see Weaknesses

**Limitations:**

Please see Weaknesses

---

> ### Author Rebuttal · Authors · 2023-08-09
>
> We address all the questions in detail as follows.
>
> >The presentation of this work is kind of confusing, making it difficult to fully comprehend the content. Specifically, in Figure 3, it is unclear what the generator loss represents if it is computed on the embedding vector. Additionally, it is unclear why the generator and regularization loss can enhance the discriminative power in Equation 10. Clarifying these help audience better understand.
>
> We extend our gratitude to the reviewer for constructive suggestions on the presentation of our work, based on which careful clarifications will be made in the revised manuscript. We summarise below the key points.
>
> (1) When the embedding map $F$ is injective, it is proved that $F(X) \neq F(Y)$ if and only if $X \neq Y$ (both $\neq$ understood in terms of distributions). Thus the generator loss, defined as the EPCFD between the embedding vectors, can be regarded as a distance function on the original time series distributions via the injective embedding map, hence preserving the discriminative power.
>
> (2) The regularization loss, defined as the EPCFD between the noise and the embedding of true time series, is proposed to enforce the injectivity of the embedding map $F_{\theta_f}$. When the embedding map and the generator $G_{\theta_g}$ are pseudo-inverses of each other and true & fake distributions coincide, the regularization loss attains zero. This justifies the use of regularization loss in Equation (8).
>
> (3) The improved discriminative power is manifested via the increased distance score between any pair of given distributions. This is ensured by maximizing the EPCFD used in generative loss and regularization loss through optimizing the model parameters $\theta_M$ and $\theta_{M}'$, respectively (see equation (10)). A close analogy in the literature is the optimization of model parameters of the test function (critic) by maximizing the Wasserstein distance in WGANs.
>
> (4) In addition, numerical evidence has been provided in Section B.4.2. Figure 6 demonstrates that the optimized $\theta_M$ increases the discriminative power of EPCFD on the stochastic process. We will include further modification in Section 3.3 in the revised manuscript.
>
> >The related work section consists of only one paragraph, which may overlook some relevant research in the field.
>
> Please refer to the *Related work* section in *Reply to all reviewers*.
>
>  >The ablation study of every component is required to demonstrate the effectiveness individually.
>
>  We provided the ablation results in Section D.1, supplementary materials in the original submission. Table 3 shows that by incorporating the regularisation loss $L_{\text{regularisation}}$ and the reconstruction loss $L_{\text{recovery}}$, the model performance has been improved ---  in terms of all three different test metrics --- consistently across empirical data. Moreover, the additional losses are proven to be more effective when applied to high-dimensional time series.
>
>
> >Regarding the sequential data, I am wondering how does this method work well on more challenging domains like video data since videos are also sequential in nature.
>
> See the *Choice of dataset* section in *Reply to all reviewers*.

---

### Official Review · Reviewer_EqQ9 · 2023-07-10

**Soundness:** 3 good
**Presentation:** 3 good
**Contribution:** 3 good
**Rating:** 7
**Confidence:** 2

**Summary:**

The paper looks at generative models for times series data. It proposes a new GAN method, based on a novel discriminator. The path characteristic function (PFC) is used as a representation of the time series distribution. Using this, a distance between two distributions is defined (PCFD) as well as a way to approximate it (EPCFD). Afterwards, it is shown how EPCFD can be used as a discriminator in a GAN scenario (resulting in the PCF-GAN). To scale it to larger input dimensions, the authors follow previous literature and introduce a parameterised dimensionality reduction of the inputs and introduce different losses in an effort to ensure that it behaves as expected. Finally, they conduct a number of experiments, showing that PCF-GAN compares favourably to competitive time-series GANs.

**Strengths:**

The method and its different components was well motivated and presented.
The paper presents a compelling theoretical motivation for the method, giving evidence on why it might be preferable to alternatives s.a. WGAN.
The experimental results are well analysed and presented.


**Weaknesses:**

The reconstruction functionality is not motivated, nor explained. It can be unclear to the unfamiliar reader why one would want it, given that we can just generate images.

Related work - similar work is only lightly touched upon and the advantages/disadvantages are not outlined, making it harder to compare.

The loss function in Eq. 9 is not arrived at in a principled way and has two hyper-parameters which might be difficult to tune?

In the experiments, the dimensionality of the inputs, as well as the length of the sequences appears rather low. This suggests potential scalability issues?



**Questions:**

In Eq. 9, do you expect training only on “ -L_recovery - L_regularisation”  to perform much worse?

**Limitations:**

The method has computational limitations which limits the number of dimensions which it can be applied to. As mentioned before, an embedding functions is trained to reduce the dimensionality of the inputs, however, if the effective dimensionality is high, the method would fail.

---

> ### Author Rebuttal · Authors · 2023-08-09
>
> We address all the questions in detail as follows.
>
> >The reconstruction functionality is not motivated, nor explained. It can be unclear to the unfamiliar reader why one would want it, given that we can just generate images.
>
> Time series reconstruction potentially has broad applications in privacy preservation [1] and in extracting semantical representation for downstream tasks via latent embedding
> [2]. The reconstruction functionality allows us to share the trained generative model and the embedding of real data as the surrogate of actual data. The separation of model and embedding reduces the risk associated with data sharing while still permitting high-quality reconstruction. We will add this to the introduction in the revised version.
>
> >Related work - similar work is only lightly touched upon and the advantages/disadvantages are not outlined, making it harder to compare.
>
> Refer to the *Reply to all reviewers*.
>
> >The loss function in Eq. 9 is not arrived at in a principled way and has two hyper-parameters which might be difficult to tune?
>
> The loss function in Eq. (9) incorporates two additional terms $L_{{\rm recovery}}$ and $L_{{\rm regularisation}}$ as penalty components. They are used to encourage the constraint $L_{{\rm recovery}}=0$ and $L_{{\rm regularisation}}=0$ for preserving injectity of the embedding $F_{\theta_f}$. The inclusion of penalty terms to transform constrained optimisations into unconstrained ones is a standard technique, exemplified in Lasso regression and Ridge regression. Empirically, the values of two hyperparameters $\lambda_1$ and $\lambda_2$ can affect the training process and should be carefully chosen via grid search. Heuristically, the optimal $\lambda_1$, $\lambda_2$ are chosen when the magnitude of three losses are comparable.
>
>
> >In the experiments, the dimensionality of the inputs, as well as the length of the sequences appears rather low. This suggests potential scalability issues?
>
> In our experiments, the time dimension of RV dataset could be as high as 200, whereas the path dimension of EGG is 14 (at least moderate or high). More importantly, the computational complexity of the PCFD used in our PCF-GAN is linear in  time and dimensions, so the scalability issue is not expected for high-dimensional time series. Specifically, let $\{T,d\}$ and $m$ be  $\{time, path\}$ dimension of time series and matrix order of EPCFD, resp. Computation of PCFD loss $\sim \mathcal{O}(Tdm^2)$, which is linear in both time and storage, hence is scalable from the computational perspective. The choice of hyperparameter $m$ may not be directly related to $T$ and $d$. With the embedding layer incorporated, the PCFD loss computation is linear in output dimension of the embedding layer. Please see our reply to **Limitations** section for additional comments.
>
> Questions:
>
> >In Eq. 9, do you expect training only on “ -L\_recovery - L\_regularisation” to perform much worse?
>
> Yes, training solely on $-L_{\rm recovery} - L_{\rm regularisation}$ leads to significantly worse performance. These two components only ensure the injectivity of the embedding $F$, but are unable to match $F(G(Z))$ and $F(X)$ without including $L_{\rm generator}$. We ran a simple experiment with only $-L_{\rm recovery} - L_{\rm regularisation}$; the training completely failed from the beginning.
>
> Limitations:
>
> >The method has computational limitations which limits the number of dimensions which it can be applied to. As mentioned before, an embedding functions is trained to reduce the dimensionality of the inputs, however, if the effective dimensionality is high, the method would fail.
>
> Our model is scalable with respect to the dimension of embedding output. However, in the case that a large matrix size $m$ is essential for the training of some high-fidelity generative model, the current PCF-GAN might experience computational bottlenecks. In this regard, incorporating the embedding layer effectively reduces the need for large $m$ (see numerical examples in our paper). It is also worthwhile to investigate low-rank approximation of relevant matrices or combine other advanced GAN model architectures to enhance the scalability of our PCF-GAN. We will add a brief discussion on it to "Limitation and Future work" section.
>
>
> [1] Rastogi, Vibhor, and Suman Nath. "Differentially private aggregation of distributed time-series with transformation and encryption." In Proceedings of the 2010 ACM SIGMOD International Conference on Management of data, pp. 735-746. 2010.
>
> [2] Cho, Kyunghyun, Bart Van Merriënboer, Caglar Gulcehre, Dzmitry Bahdanau, Fethi Bougares, Holger Schwenk, and Yoshua Bengio. "Learning phrase representations using RNN encoder-decoder for statistical machine translation." arXiv preprint arXiv:1406.1078 (2014).

---

### Official Review · Reviewer_EJ4R · 2023-07-10

**Soundness:** 3 good
**Presentation:** 3 good
**Contribution:** 3 good
**Rating:** 5
**Confidence:** 2

**Summary:**

The paper proposes an approach to improve time-series modeling in Generative Adversarial Network framework by using path characteristic function as the embedding of the time series sample. It explains the feasibility of PCF distance and how to integrates it as a distance measure between two time series data in order to help the discriminator learn better generative features. The goal and benefit of the proposed method is to empirically improve the generation quality of time series data.


**Strengths:**

The paper explores using a novel loss function in Generative Adversarial Network framework. It computes the proposed empirical path characteristic function distance between generated and given sample to learn better time series generative features. The proposed method outperforms other generative baselines on the given datasets. The paper is well written ( and includes the codebase in appendix for reproducibility).


**Weaknesses:**

It would be great to have to following experiments in order to compare fairly with the existing baselines:

1. Comparison with existing baselines like COT-GAN on metrics like
(a) the sum of the absolute difference of the correlation coefficients between channels avg over time
(b) absolute difference between the correlation coefficients of real and generated samples.

2. Comparison on high dimensional dataset (like Sprites and human action sequences with FID, KID scores )


**Questions:**

N/A

**Limitations:**

As mentioned in the weakness section, having more ablations and experiments for comparison would help the reader better understand the efficacy of the proposed approach.

---

> ### Author Rebuttal · Authors · 2023-08-09
>
> We address all the questions in detail as follows.
>
> >Comparison with existing baselines like COT-GAN on metrics like (a) the sum of the absolute difference of the correlation coefficients between channels avg over time (b) absolute difference between the correlation coefficients of real and generated samples.
>
>
> We have included the two metrics on (a) auto-correlation and (b) cross-correlation with different lags as additional test metrics to evaluate the quality of the generative models. Please see the *test metrics* section in 'Reply to all reviewers'.
>
>
> >Comparison on high dimensional dataset (like Sprites and human action sequences with FID, KID scores)
>
> We refer to the discussion on the *choice of datasets* section in *Reply to all reviewers*.
>
> *Ablation study*
>
> Regarding the comments in the Limitations section, the ablation results have been provided in Section D.1 of supplementary materials in the original submission. Table 3 shows that by incorporating the regularisation loss $L_{{\rm regularisation}}$ and the reconstruction loss $L_{{\rm recovery}}$, the model performance has been improved --- in terms of all the three different test metrics --- consistently across empirical data. Moreover, the additional losses are proven to be more effective when applied to high-dimensional time series.

---

> > ### Comment · Reviewer_EJ4R · 2023-08-21
> >
> > Dear Authors,
> > Thanks a lot for addressing my concerns and providing the quantitative comparison with other baselines.

---

> > > ### Author Response · Authors · 2023-08-21
> > >
> > > Thank you very much for your confirmation!

---

### Official Review · Reviewer_BgyV · 2023-07-10

**Soundness:** 3 good
**Presentation:** 2 fair
**Contribution:** 3 good
**Rating:** 5
**Confidence:** 4

**Summary:**

This paper proposes a path characteristic function GAN, named PCF-GAN, for learning to generate time-series data. More specifically, the authors mainly employ the rough path theory to build the PCF distance, such that the temporal cues can be encoded by unitary features, enabling the PCF to learn sequential data. The PCF distance is then proved to be complete and favorable for minimizing the difference between two stochastic processes. Even though, an encoder-decoder-based embedding is proposed to compare two random processes in the latent space of reduced dimensions, catering for the ease of training in practice. Experimental results have also verified the effectiveness of the proposed PCF-GAN.

**Strengths:**

The established theory on PCF is novel and complete, by far my understanding. The authors also propose a practical way of minimizing the PCF distance, by using an encoder-like embedding that is able to reconstruct signals.

**Weaknesses:**

1. I am appreciated the theory behind the PCF. However, the usage of path theory needs to be clarified, when aiming at improving characteristic functions to represent stochastic process. The authors claim that CF-metric fails to capture temporal dependency of sequential data, which is true. However, why using path theory on characteristic functions can well address the temporal dependency, given the fact that there are other representations of the characteristic function of a stochastic integral?

2. The authors claim that the embedding operation does not have any gradient constraints. However, when adding calculating the EPCFD after the embedding layers, the continuous and differential properties may not hold, without the restriction on the embedding layers. I am not sure whether the reconstruction regularization can compensate strictly. The authors are encouraged to elaborate more on this.

3. Another weakness is regarding the experimental validations, which were performed based on low-dimensional time-series datasets. The PCF-GAN is shown to beat the COT-GAN on these datasets. However, COT-GAN is also able to generate short video clips, in which the scenarios are much complicated. Would the PCF-GAN scale well to the higher-dimensional generation tasks, such as short video generation? Also the comparing baselines are not the state-of-the-art. It is the fact that many recent methods have been reported to surpass COT-GAN and TimeGAN, for example, [1] and [2].

[1] Seyfi, Ali, Jean-Francois Rajotte, and Raymond Ng. "Generating multivariate time series with COmmon Source CoordInated GAN (COSCI-GAN)." Advances in Neural Information Processing Systems 35 (2022): 32777-32788.
[2] Jarrett, Daniel, Ioana Bica, and Mihaela van der Schaar. "Time-series generation by contrastive imitation." Advances in Neural Information Processing Systems 34 (2021): 28968-28982.



**Questions:**

1. In Equation (4), why x_0=I_m. Wouldn't it be y_0=I_m for the differential equation?

2. In Line 254, what does it mean by IGM? Should it be IPM?

3. Why for the regularization loss, this paper employs a different \theta'_M? How \theta_M and \theta'_M were implemented in this paper?

**Limitations:**

Please see my weakness.

---

> ### Author Rebuttal · Authors · 2023-08-09
>
> We address all the questions in detail as follows.
>
> **1. Temporal dependency**
>
> The path characteristic function (PCF) is proven to characterize the law of the stochastic processes, or time series. Hence, PCF completely determines the temporal dependency, *e.g.*, statistics like auto-correlation. The fundamental mechanism lies in the non-commutativity of matrix multiplication --- the multiplication in the Lie groups used to define PCF --- which echoes the non-interchangeability of the temporal order of events in time series. Note that the PCF deals with the measures on the *$\infty$-dimensional* space of paths, while the usual characteristic function (CF) is defined for $\mathbb{R}^d$-valued random variables. The latter cannot be directly extended to the $\infty$-dimensional setting mainly because the range of the exponential map used in CF is $i \mathbb{R}$, which is commutative.
>
> There might be other representations characterizing the law on the path space; however, to the best of our knowledge, the PCF is the only such representation that has been proven characteristic, uniformly bounded, and differentiable.
>
>
> **2. Embedding layers**
>
> We thank the reviewer for pointing out the potential restrictions caused by embedding layers. Some clarifications and further explanations are as follows:
>
> 1. We concur with the reviewer on the need for the differentiability assumption of the embedding layer. As long as the embedding map is differentiable, the continuity and differentiability properties (Theorem 3.6) hold after applying the embedding layers. In fact, it has been taken for granted that most of neural networks used for the embedding map are differentiable, so as to ensure the validity of gradient descent algorithms. We shall clarify this point in the revised manuscript.
>
> 2. Indeed, gradient constraints are required by various GANs models, such as Wasserstein GANs, in order to impose additional regularity or add gradient norm upper bound on the critic function/embedding layer. In these GAN models, the distance may become unbounded in absence of further constraints. However, the PCFD distance between any two distributions on the path space is *uniformly* bounded by $2m^2$, as proved in Lemma 3.5. Therefore, when applying EPCFD on the embedded distributions, the uniform boundedness assumption holds automatically.
>
> 3. In our work, the reconstruction and regularization losses are not introduced to address the issues of gradient constraints,  continuity, or differentiability --- these have been taken care of naturally by the aforementioned mathematical properties of PCFD. Instead, the reconstruction and regularization losses aim at imposing injectivity on the embedding function $f$, so that  $X \stackrel{d}{=} Y$  whenever $f(X) \stackrel{d}{=} f(Y) $ for any path-valued random variables $X$ and $Y$ (here $\stackrel{d}{=}$ means equality in distributions). The following trivial example validates the assumption on the injectivity on $f$: if $f$ is simply a constant function, then no matter whether distributions of $X$ and $Y$ agree or not, one has $\mathrm{PCFD}(f(X),f(Y)) = 0$.
>
> **3. Experimental validations**
>
> To address this question, we first refer to the discussions on comparison with existing literature and choice of datasets in *Reply to all reviewers*. Besides, in what follows let us elaborate on our method in the context of Refs. [2, 3].
>
> The main objective of this work is to propose a novel distance metric to improve the quality of GANs for time series generation. In contrast, both [2] and [3] proposed new frameworks for generating sequential data, which may not be suitable to benchmark directly with our method. But it would be interesting to incorporate our methods within their framework. Specifically, [2] proposed a framework to tackle multivariate time series by comparing each channel of the real and fake times series and then comparing across all channels globally. Within this framework, PCFD can be used as a distance measure --- in place of the average cross-entropy loss across time --- for both channel-wise and global comparisons. Similarly, for [3] we may replace the average cross-entropy loss by PCFD in their min-max objective function (see Equation (11) in [3]).
>
> Questions:
>
> **1.** Yes, this is a typo. We will fix it accordingly.
>
>
> **2.** IGM stands for Implicit Generative Model. It was our mistake for omission on its clarification. We will modify this part accordingly.
>
> **3.** The generator loss is the EPCFD distance between the reconstructed noise $F_{\theta_f}(G_{\theta_g}(Z))$ and $F_{\theta_f}(X)$. In contrast, the regularization loss is the EPCFD distance between the noise $Z$ and $F_{\theta_f}(X)$. Although the reconstructed noise converges to the noise distribution eventually, there is no guarantee that they are the same throughout the optimization process. Thus these two losses may have distinctive trainable parameters.
>
> We initialize $\theta_M$ and $\theta_M'$ following the method described in the supplementary material, Section B.4.1 in the original submission. The optimization for $\theta_M$ follows the procedure in [4]. Note that the parameter $\theta'_M$ has been initialized and optimized independently of $\theta_M$.
>
> [1] Chevyrev, Ilya, and Terry Lyons. "Characteristic functions of measures on geometric rough paths." (2016): 4049-4082.;
>
>
> [2] Seyfi, Ali, Jean-Francois Rajotte, and Raymond Ng. "Generating multivariate time series with COmmon Source CoordInated GAN (COSCI-GAN)." Advances in Neural Information Processing Systems 35 (2022): 32777-32788.
>
> [3] Jarrett, Daniel, Ioana Bica, and Mihaela van der Schaar. "Time-series generation by contrastive imitation." Advances in Neural Information Processing Systems 34 (2021): 28968-28982.
>
> [4] Lou, Hang, Siran Li, and Hao Ni. "Path Development Network with Finite-dimensional Lie Group Representation." arXiv preprint arXiv:2204.00740 (2022).

---

> > ### Comment · Reviewer_BgyV · 2023-08-21
> >
> > The authors rebuttal has addressed my concerns on the rationale on using PCF, together with the associated gradient constrains. The authors are also encouraged to indicate whether they used any gradient penalty or spectrum norm in practice. For the experiments, I still feel quite weak evaluations on the proposed method. I am keeping my score.

---

> > > ### Author Response · Authors · 2023-08-21
> > >
> > > Thank you very much for your prompt reply. We are pleased to hear that we addressed your questions regarding the methodological aspect of our work. We don't use any gradient penalty or spectrum norm in our proposed PCF-GAN in numerical experiments. This is underpinned by the theoretical properties of the PCF-GAN. With regard to evaluation, the time constraint of the discussion period prevents us to conduct thorough numerical experiments on video data. Nevertheless, We want to draw your attention that the primary objective of our work is to introduce a novel and principle discriminator for time series generation, backed up by theoretical properties along with the empirical efficacy shown in proper benchmarking results. This PCF discriminator can be flexibly integrated with a variety of GAN frameworks, offering the potential to achieve state-of-the-art results on more challenging real-world time series data.

---

### Official Review · Reviewer_cYLd · 2023-07-27

**Soundness:** 3 good
**Presentation:** 3 good
**Contribution:** 3 good
**Rating:** 5
**Confidence:** 3

**Summary:**

The paper presents a new metric for the distributions on the path space via PCF and provides theoretical proofs for analytic properties of the proposed loss metric which benefits GAN training. It introduces a novel PCF-GAN to generate & reconstruct time series simultaneously. It compares the proposed method with prior work and shows improved performance.

**Strengths:**

- The paper is relatively well-written and the proposed approach is described clearly.

- The authors provide theoretical grounding for their work. They prove PCF's characteristicity, boundedness, differentiability with respect to generator parameters, and its weak continuity which ensures the stability and feasibility of training the PCF-GAN.

- Experimental results demonstrate improved performance compared to several prior works. The authors provide comparisons with TimeGAN, CotGAN and RGAN on RV, Stock, Air and EEG datasets using discriminative and predictive metrics.

**Weaknesses:**

- There are concerns regarding missing comparisons with related work. [A, B] also propose generative models for time series generation. [A] combines the adversarial training of GANs and the exact maximum likelihood training of CTFPs into a single framework. It designs an invertible generator, and adopt an autoencoder, on whose hidden space of the GAN performs the adversarial training. [B] uses Deep Euler representation and Wasserstein distances to propose three generative methods for times series. Two generative method EWGAN and EDGAN demonstrate an accuracy similar to state-of-the-art GAN generators and show better performance for capturing temporal dynamic metrics of the time series. The third method CEGEN is based on a loss metric computed on the conditional distributions of the time series. These papers are not mentioned and no comparisons are provided with them.

- There are metrics such as FID in prior work, e.g. [B], which are not used in the experiments.

[A] GT-GAN: General Purpose Time Series Synthesis with Generative Adversarial Networks; Jeon et al.;

[B] Conditional Loss and Deep Euler Scheme for Time Series Generation; Remlinger et al.;

**Questions:**

The authors need to clearly distinguish their work with existing literature and provide comparisons with prior work (e.g. [A, B]).

**Limitations:**

The authors have adequately discussed limitations of their work.

---

> ### Author Rebuttal · Authors · 2023-08-09
>
> We address all the questions in detail as follows.
>
> **Comparisons with related work**
>
> We first refer to the comparison to the existing literature in *Reply to all reviewers*. Then we further elaborate on the comparison between our method and [A, B].
>
> [A] proposed a GAN framework to generate time series that is robust to missing data, making extensive use of continuous-time process modules therein. In particular, their framework incorporates CFTP as generator, NCDE as the bottleneck architecture for auto-encoder and critic function, and average cross-entropy loss over time as objective function. In contrast, our work aims to improve GAN's training by introducing and incorporating  PCFD, a novel discriminator. Thus, in view of the essential differences in the architectures of generator and critic, we think it is not suitable to directly benchmark our model against the framework in [A]. We do acknowledge, however, that it is a very interesting direction to incorporate PCFD into the GAN framework in [A]. For instance, one may replace the discriminator in [A] by our proposed PCFD. As PCF relies on the unitary feature arising from controlled differential equations (*e.g.*, NCDE; see [C]), we conjecture that this operation will enhance the model's robustness to missing data, hence improving the performance compared to the traditional cross-entropy loss used in [A].
>
> [B] used the Euler discretization scheme of SDEs as a generator and the Wasserstein metrics as an objective function. For a fair comparison, one should compare PCFD in our work to the Wasserstein metrics in [B]. The use of Wasserstein metrics typically requires a 1-Lipschitz critic function, which entails further constraints on the gradient of the critic function. In contrast, in our work, further constraints are not needed at all, thanks to the boundedness of PCFD.  It would be very interesting to numerically benchmark our method to [B] and analyze the performance of the integrated model based on SDE Euler discretization and PCFD. But as their codes are not publicly available and re-implementation of their model from scratch within a week appears unfeasible, at the moment this is beyond our scope. One minor point: by [B], two generative methods EWGAN and EDGAN (besides CEGEN) consistently underperform against COT-GAN.
>
> An important innovation of CEGAN is the use of conditional loss, which leads to superior performance ([B]). It would be interesting to substitute the $W_2$ metric by PCFD therein to match the distribution of $\mathbb{P}[X_{t+1} | X_{t}]$. More importantly, this enables us to extend the loss to match the conditional law of Step $1$ and Step $q$, *i.e.*, $\mathbb{P}[X_{t+1: t+q} | X_{t}]$. We will briefly mention it in the future work section.
>
> **Additional evaluation metrics**
>
> The FID score, commonly used in image generation, requires a pre-trained model to map each sample to a high-dimensional vector. For time series data, the choice of appropriate outputs or labels for pre-trained models encompasses ambiguity. In addition, the FID score requires the assumption of normality to avoid bias ([D]). Meanwhile, in our humble opinion, FID is not widely used in time series generation literature (*cf.* works on GT-GAN [A], TimeGAN, and COT-GAN). Taking the above into consideration, together with the fact that [B] did not provide implementation details or code for the FID score, we decide not to include this metric in our experiments.
>
> Instead, we consider three additional metrics: auto-correlation, cross-correlation and marginal distribution metrics. They are intended to assess the temporal dependency, spatial dependency, and marginal fitting via classical statistics, respectively. See the discussion on the evaluation section in *Reply to all reviewers* and the attached supplementary PDF.
>
> [A] Jeon, Jinsung, Jeonghak Kim, Haryong Song, Seunghyeon Cho, and Noseong Park. "GT-GAN: General Purpose Time Series Synthesis with Generative Adversarial Networks." Advances in Neural Information Processing Systems 35 (2022): 36999-37010.
>
> [B] Remlinger, Carl, Joseph Mikael, and Romuald Elie. "Conditional loss and deep Euler scheme for time series generation." Proceedings of the AAAI Conference on Artificial Intelligence. Vol. 36. No. 7. 2022.
>
> [C] Lou, Hang, Siran Li, and Hao Ni. "Path Development Network with Finite-dimensional Lie Group Representation." arXiv preprint arXiv:2204.00740 (2022).
>
> [D] Heusel, Martin, Hubert Ramsauer, Thomas Unterthiner, Bernhard Nessler, and Sepp Hochreiter. "GANs trained by a two time-scale update rule converge to a local Nash equilibrium." Advances in Neural Information Processing Systems 30 (2017).

---

### Author Rebuttal · Authors · 2023-08-09

We thank all the reviewers for their insightful comments and constructive suggestions. We are pleased that all the reviewers find our work novel, sound, and theoretically motivated. We also acknowledge the shared questions from the reviewers on related work and numerical evaluation.

**Comparison to existing literature**:

 - Related work: We will extend the related work section in the revised version by adding discussions on other state-of-the-art time series methods, including the comparison with our model.  We plan to incorporate additional GAN models with autoencoders (*e.g.*, GT-GAN). We shall also delve into strategies for improving GAN training for time series, highlighting innovative approaches such as additional conditional loss and SDE-based neural networks (Deep Euler representation [2]).

-  Baselines: A main contribution of our PCF-GAN lies in the innovation of the discriminator for time series generation,  built upon the principled mathematical representation PCF. To validate its effectiveness, we focus on those state-of-the-art models with discriminators tailored for time series distribution. We thus choose RCGAN and TimeGAN (which use the average cross-entropy loss over time as the discriminator) as well as COTGAN (whose discriminator loss based on causal optimal transport). For a fair comparison, we use the same generator across all GAN models in our numerical experiments.

    We acknowledge that several other time series generative models (GT-GAN, COSCI-GAN and EWGAN) pointed out by the reviewers may have superior performance, on certain benchmarking datasets, than the baselines adopted in our paper. Nevertheless, these models focus mainly on network framework and generator architecture, while our work emphasises the innovation of a discriminator. Thus, benchmarking our method with [1, 2, 3] as suggested by **cYLd** and **BgyV** appears not directly applicable.

- Future work. Although our numerical experiments do not include the comparison with the additional baselines, it is certainly of great interest to explore how PCFD could be incorporated into these more recent and state-of-the-art GANs for time series generation as future work.  Indeed, as a distance metric on time series, PCFD can be flexibly incorporated with other advanced generators of time series GAN models, hence may further improve the performance. For example, one can replace the average cross-entropy loss used in [1, 3] and the Wasserstein distance in [2] by PCFD, with some simple modifications on the discriminators. Such considerations will be elaborated in the future work section,  highlighting the flexibility of our proposed PCF-GAN.

**Evaluation**:

- Test metrics. In response to the reviewers' suggestions, we have included several additional commonly-used test metrics to evaluate the quality of the generative models from different perspectives. (1) Fitting of temporal dependency. We adopt auto-correlation metrics used in COT-GAN and cross-correlation metrics used in Quant-GAN, as suggested by Reviewer EJ2R. These are classical statistical tools to measure the temporal dependency of time series within and across channels. (2) Fitting of marginal distribution. Here we adopt the distributional metric in Quant GANs [4]. We benchmark our method against RGAN, COT-GAN, and TimeGAN. Results in the supplementary PDF show that PCF-GAN significantly outperforms baselines. We will include the additional results of these test metrics in the appendix of the revised manuscript.

- Choice of datasets. To demonstrate the effectiveness of our proposed method, we have benchmarked several sequential datasets from different domains with various characteristics, as summarised in Table 1.  We acknowledge that it is of interest to validate our methods on more complex sequential data, such as video and human action. Nonetheless, training generative models on these tasks often requires tailored network architecture (*e.g.*, using deep convolution modules to learn spatial dependency within each frame of video) or more advanced frameworks, which lies out of the main focus of this paper. Furthermore, to our knowledge, these datasets are not commonly used for benchmarking generative models for sequential data (with the exception of COT-GAN, which included a short clip of video data). That said, it is certainly an intriguing direction to incorporate PCFD into complex model architectures/frameworks to tackle more intricate sequential data. This will be addressed in the future work section of the revised manuscript.

[1] Jeon, Jinsung, Jeonghak Kim, Haryong Song, Seunghyeon Cho, and Noseong Park. "GT-GAN: General Purpose Time Series Synthesis with Generative Adversarial Networks." Advances in Neural Information Processing Systems 35 (2022): 36999-37010.

[2] Remlinger, Carl, Joseph Mikael, and Romuald Elie. "Conditional loss and deep Euler scheme for time series generation." Proceedings of the AAAI Conference on Artificial Intelligence. Vol. 36. No. 7. 2022.

[3] Seyfi, Ali, Jean-Francois Rajotte, and Raymond Ng. "Generating multivariate time series with COmmon Source CoordInated GAN (COSCI-GAN)." Advances in Neural Information Processing Systems 35 (2022): 32777-32788.

[4] Wiese, Magnus, Robert Knobloch, Ralf Korn, and Peter Kretschmer. "Quant GANs: deep generation of financial time series." Quantitative Finance 20, no. 9 (2020): 1419-1440.

---

> ### Author Response · Authors · 2023-08-20
>
> Thank you foryour constructive and insightful feedback again. We would like to confirm if our response has adequately addressed your concerns. As the discussion deadline is approaching, we are keen to hear your comments. Please don't hesitate to pose any further questions you might have. We will address these questions promptely.

---

### Decision · Program_Chairs · 2023-09-21

**Decision:**

Accept (poster)

**Comment:**

This work addresses generative models of time series, proposing a new metric on time series, PCFD, which they successfully apply to the discriminator of GANs. The presentation is good, well justified and motivated, and the empirical analysis well-validates the approach. There were some concerns about scaling, but the author's seem to have been able to argue successfully that the evaluations are sufficient for the story. Overall, the discussions had a strong positive influence on the final form (e.g., see author notes below), so I recommend this paper is accepted as a poster.